# Global green hydrogen-based steel opportunities surrounding high quality renewable energy and iron ore deposits

Alexandra Devlin ●[1], Jannik Kossen[2], Haulwen Goldie-Jones[1] & Aidong Yang ●[1] ✉

The steel sector currently accounts for 7% of global energy-related $CO_2$ emissions and requires deep reform to disconnect from fossil fuels. Here, we investigate the market competitiveness of one of the widely considered decarbonisation routes for primary steel production: green hydrogen-based direct reduction of iron ore followed by electric arc furnace steelmaking. Through analysing over 300 locations by combined use of optimisation and machine learning, we show that competitive renewables-based steel production is located nearby the tropic of Capricorn and Cancer, characterised by superior solar with supplementary onshore wind, in addition to high-quality iron ore and low steelworker wages. If coking coal prices remain high, fossil-free steel could attain competitiveness in favourable locations from 2030, further improving towards 2050. Large-scale implementation requires attention to the abundance of suitable iron ore and other resources such as land and water, technical challenges associated with direct reduction, and future supply chain configuration.

At present, fossil fuels are the steel sector's bloodstream: 27 EJ ($10^{18}$ J) of coal, 3 EJ of gas and 5 EJ (1400 TWh) of electricity are consumed annually for the production of the mostly widely used metal on earth[1], emitting an average of 2 tonnes of $CO_2$ per tonne of steel and causing 7% of global energy-related $CO_2$ emissions[2]. 1.95 billion tonnes of steel were produced in 2021[3], with a projection to increase to 2.19 billion tonnes by 2050 given global demand converging to 250 kg per capita in 2080[4]. Currently, 22% of steel production is via the secondary (scrap-based) electric arc furnace (EAF) route which is set to rise to up to 50% of demand by 2050 as projected by Pauliuk, et al.[5], so long as effective scrap collection, contaminant control, and trade is sustained. Exhaustive material efficiency measures of steel-containing products, including enhanced durability, reusability, and minimalist design, could reduce primary (ore-based) steel demand, potentially by up to 40%[6]. Global economic advancement and population growth, however, counteract against the prospects of steel demand reduction; emission forecasts are calling for urgent joint supply- and demand-side mitigation measures[7]. A large segment of future steel demand will likely need to be met by primary steel, during which emission-intensive

carbon-based iron ore reduction would occur if the use of the current technology is continued.

Responding to the pressure for decarbonisation, incremental measures such as improving the energy efficiency and partial fuel switching (biomass or hydrogen) of fossil-based operations will be insufficient to meet the steel sector's climate commitments; the blast furnace must be retrofitted with carbon capture technology, or phased out[8]. On the other hand, deep decarbonisation technology has emerged in varying scales of emissions abatement, technical feasibility, economic viability, and development maturity. Whilst electric steelmaking furnaces can be readily decarbonised through renewable power, the most promising options to decarbonise ironmaking are: (i) green hydrogen($H_2$)-based direct reduction of iron (DRI), (ii) natural gas (NG)-based DRI with carbon capture, utilisation and/or storage (CCUS), (iii) traditional blast furnace (BF) or smelting reduction (SR) with partial substitution of coal with biomass and CCUS, and (iv) direct iron ore electrolysis[9–11]. $CO_2$ capture solutions have so far had very limited success in the steel sector; only one NG-based DR plant operates with CCUS[12]. Retrofitting existing BF plants with CCUS, despite

[1]Department of Engineering Science, University of Oxford, Parks Road, Oxford OX1 3PJ, UK. [2]OATML, Department of Computer Science, University of Oxford, Parks Road, Oxford OX1 3PJ, UK. ✉e-mail: aidong.yang@eng.ox.ac.uk

being desirable due to use of existing assets, has not yet been trialled nor is an effective emission abatement method given the plurality of emission points and variability in $CO_2$ concentration of flue gases[13]. Representing a completely different direction, both $H_2$-DRI and electrowinning are renewable energy-based solutions where carbon as the reducing agent is completely replaced by hydrogen or electricity, respectively. As a revolutionary technology, electrowinning is currently cost-prohibitive and expected to reach commercial-readiness in the long-term (post 2040)[14]. In comparison, $H_2$-DRI combined with the electric arc furnace (EAF) (termed $H_2$-DRI-EAF) has widely been regarded as a leading deep decarbonisation option despite a range of issues to be addressed[15], thanks to the intensifying industrial investments[16], successful pilot by Swedish forerunners[17] and planned commercial production by 2025[18].

Fossil-based Direct Reduction (DR) (using natural gas or gasified coal) is already familiar to industry, with global DRI production reaching 120 Mt in 2021. Despite a 14% DRI capacity increase from the previous year, global steel production share from the DRI-EAF route was only about 7%[3,19]. Hence, not only the technical difficulty of switching fuels to green-$H_2$, but also the economic challenge of dramatic capacity expansion, must be overcome for a successful switch to the near-zero carbon $H_2$-DRI-EAF route. Vast quantities of renewable energy (RE), the supply of which is often subject to variability and intermittency, will be required to electrolyse water for $H_2$ production and to power the EAF. In addition, for the DR technology with greatest industrial diffusion, that is the MIDREX or HYL-Energiron shaft furance[20], iron ore supply must adhere to strict quality requirements to preserve steel quality and iron- and steelmaking productivity[15]. The shaft furnace operates using a counterflowing ore and reducing gas to produce a sponge iron product, requiring ore in pellet form with minimum 67% Fe content[21] to limit ore impurities and prevent downstream (EAF) difficulties; for reference, global average extracted ore Fe content is 62%[22]. This is distinct from the 65% Fe content requirement for blast furnace production where both molten iron and slag are formed, enabling the removal of gangue and more flexible ore forms including sintered fines due to the physical 'bath' structure[21]. Beneficiation, which describes the physical and/or chemical separation processes to remove impurities (commonly silicon, aluminium, phosphorus, and sulphur) from, and thereby increase the Fe-content of, iron ore, is increasingly relied on by the iron ore industry to accommodate lower grades of mined ore[23–25]. This trend is most likely to continue in order to supply significantly up-scaled DRI production.

Despite these challenges, green $H_2$-DRI-EAF technology presents a unique opportunity to reassess the location of production facilities and consequent supply chain configurations, in order to optimise the use of locally available resources. To date, a few studies have investigated cost-minimised $H_2$-DRI-EAF production, although with limited spatial coverage: the UK[26] and Northern Europe[27]. Globally, Bataille, et al.[4] projected decarbonised steel sector pathways based on existing (fossil-based) facilities through their adoption of CCUS and renewables-based technology. Lopez, et al.[28] also investigated global energy needs of renewables-based steel production, however, without incorporating regional resource differentiation nor renewables optimisation. The common assumption was that future production facilities will correlate with locations (but not necessarily capacity) of current production facilities. Whilst existing infrastructure surrounding current steelmaking locations is advantageous to utilise, their geographical setting does not necessarily offer favourable climatic and geological resources for renewables-based production. Industrial relocation as an enabler for industrial decarbonisation has been explored by several regional case studies, including the work of Gielen, et al.[29] and Wood, et al.[30] on Australia's potential future role as a near zero-carbon steelmaker, and that of Trollip, et al.[31] on an assessment of South Africa's opportunity to supply Europe with near zero-carbon iron.

In this study, we provide a baseline assessment of the global potential of renewables-powered steel production using green $H_2$-DRI-EAF technology, where the entire supply chain is co-located at the iron ore mine vicinities. Taking into account the RE and ore supply challenges, this work particularly considers the geographical distribution of local resources as considered in an earlier regional study (focussing on Australia and Japan), which demonstrated the financial rewards of co-locating steelmaking with high-quality natural resources[32]. The key question of interest is, across all the major iron ore deposits, to what extent these locations show promising green $H_2$-based steel economics. Answering this question will identify regional opportunities and inform the future design of more sophisticated supply chains for regions which need to go beyond the simple co-location strategy, involving trade of ore, RE products, and/or intermediate steelmaking products, such as hot briquetted iron (HBI).

The main part of the study consists of two steps, as illustrated in Fig. 1. In the first step, a techno-economic optimisation model was developed for localised green $H_2$-based steel production adjacent to iron ore deposits, based on regional RE profiles at hourly temporal resolution. The optimal capacities of all key technical components (energy supply and storage, DR shaft furnace, EAF) were determined for a $H_2$-DRI-EAF steel facility with 1 Mtpa output, a reference scale chosen based on the range of scale for EAF steelmaking (0.3–3 Mtpa, Renda, et al.[33]) and current DR ironmaking nominal capacities (0.4–2.5 Mtpa, Global Energy Monitor[34]). Applied to three deployment (hence technology advancement) timelines (2030, 2040, 2050) combined with different levels of scrap steel (0%, 25%, 50%), the model was solved for 44 regions in 17 important iron ore producing countries. For each region, the cost of obtaining iron ore suitable for DRI based on the national iron ore quality data was further estimated and combined with the labour cost and the output of the optimisation model for steel production, to establish the overall cost of green $H_2$-based steel. Due to limited global ore petrology data availability, Fe content was the single indicator used for iron ore quality. Solar and wind were chosen as the core renewable energy resources for analysis as they are available in all studied locations, and are predicted to deliver 70% of global electricity generation in 2050[35]. In the second step, outputs from the optimisation model were processed using machine learning techniques to derive a green $H_2$-based steel investment model, which, thanks to its much higher computational efficiency, enabled expansion of the global assessment to >300 deposits in 68 countries.

Islanded (100%) RE supply, although ensuring nearly zero carbon emissions, exhibits intermittency and variability which can be managed using energy storage and/or oversizing flexible production plant[36, 37]. For comparison, we also assessed $H_2$-DRI-EAF plants powered by grid electricity, which offers stable energy supply. We assumed that the grid power's carbon footprint was dependent on the forecast power mix and that electricity was charged according to current industrial tariffs, for the sake of assessing conditions for competitive steelmaking and providing recommendations for electricity market reform. The economic and carbon emission perspectives of both schemes were evaluated with reference to the conventional BF-BOF route. Finally, the energy, water and land use implications and further challenges that the $H_2$-DRI-EAF route needs to address for large-scale implementation were considered.

Through combined use of optimisation and machine learning, we demonstrate in this study that for market competitiveness, it is important especially in the short term to locate flexible green-$H_2$ based steelmaking in favourable locations, commonly characterised by strong and reliable solar, fair supplementary wind, and high-quality iron ore. If coking coal prices remain high and projected cost reductions are realised for electrolysers, solar panels and wind turbines, the majority of locations could become competitive against the fossil-based BF-BOF route by 2050. By illustrating the global map of projected solar and wind-powered steelmaking costs, we show that green

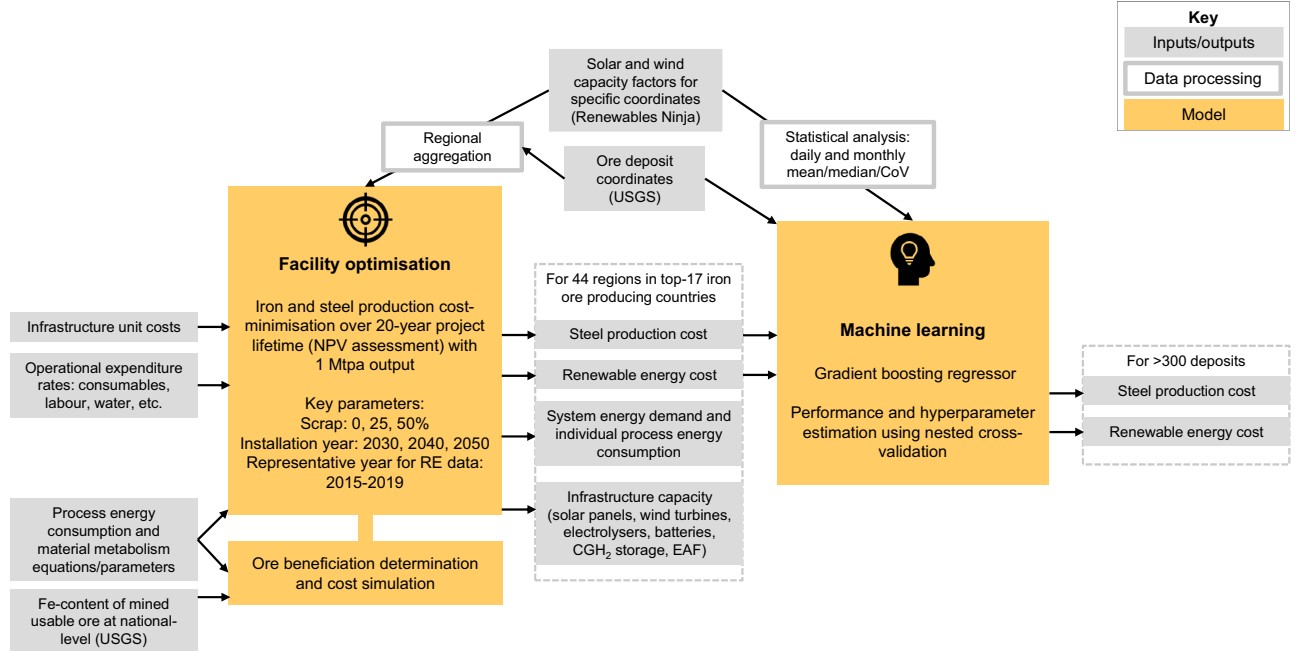

**Fig. 1 | Schematic of methodology for the global green H₂-based steel production assessment.** Facility-level optimisation was carried out for 44 regions in 17 important iron ore producing countries, the results of which were fed into a machine learning model to extend the spatial coverage to over 300 global iron ore deposits.

steel investment decisions must revolve around climatic and geospatial factors.

## Results

### High-quality renewables and ore for low-cost green H₂-based steel

Our modelling of green H₂-based steel production, powered by islanded RE systems (solar and onshore wind) and localised at the iron ore mine site, projected levelised costs of green steel (LCOS) in the range of $535-972/t without scrap charging, as illustrated in Fig. 2. By 2050, the LCOS range dropped to $535-$831/t, with LCOH₂ from $1.63 to 2.80/t and levelised costs of energy (LCOE, calculated based on energy consumed by steel making) from $16 to 50/MWh. Costs varied according to the region (affecting RE potential, iron ore quality, and workforce wages) and project installation year (affecting emerging technology unit costs and electrolyser efficiency). Favourable locations were solar-dominant in RE infrastructure albeit with some wind capacity to balance the diurnal profile; in 2050, the solar portion of RE capacity in the most favourable locations, being Iran, Peru, South Africa and Chile, were 89%, 100%, 80%, and 100%, respectively. Strong solar power potential with minimal seasonal variation allowed the production systems to profit from cheaper unit costs for solar PV panels compared to wind turbines ($327/kW and $835/kW, respectively, in 2050[38]) and reduce electrolyser oversizing and/or storage requirements. Across all cases, most energy storage costs were allocated to compressed gaseous H₂ (CGH₂) (mean 91%) with some electricity storage in batteries to manage RE variability; on average, 50% of produced H₂ was stored temporarily as CGH₂ whilst electrolysers and EAFs were oversized by factors of 2.3 and 1.3, respectively. Affordable green H₂-based steel production was correlated to lower energy- and land-intensity.

Whilst the islanded energy supply has considered only locally available wind and solar resources, which represent the RE types that are most likely to dominate the growth of RE capacity in most countries in the next few decades, contributions from continuous and controllable zero carbon energy sources, i.e. hydropower (very important for countries such as Sweden and Brazil) and nuclear may

play a critical role in competitive green H₂-based steel production. Although these alternate energy sources are not considered in the islanded case, they are reflected in the evolution of national energy mix in the grid-based case presented later.

As illustrated in the cost breakdown of Fig. 2b, renewable energy capacity was more influential than iron ore Fe-content for cost performance, yet DR-grade ore beneficiation requirements challenged profitability. For example, Russia was disadvantaged by poor quality solar and mediocre wind resources, which high-quality ore (71% Fe-content) was able to only partially offset, resulting in low market competitiveness. By sensitivity analysis, a 10% change in ore costs caused an average 3% change in LCOS across all sites. If the ore quality was too low, however, beneficiation requirements would significantly increase production costs. In Kazakhstan, where the renewable energy supply is of reasonable quality thanks to its strong wind resources (as reflected in the LCOE and LCOH₂) but where ore is mined at extremely low quality (20% Fe-content, well below the global average of 62%), beneficiation requirements were immense and the DR-grade pellet premium tripled the average (from $40/t to $122/t ore); total ore cost accounted for 45% of steel production costs, well above the 27% average. Whilst Kazakhstan represents an extreme case, in general, current economic ore resources serve the BF market and may prove unprofitable for the DR iron-making market with enhanced quality requirements.

### Market competitiveness achievable in the near term for ideal locations

The green H₂-DRI-EAF production route may be cost-competitive with the BF-BOF in favourable locations in the next decade, and for most locations towards 2050. The projected costs for green-hydrogen based steel produced at iron ore mine sites are plotted against current BF-BOF costs in Fig. 3a. BF-BOF operational costs in 2021 ranged from $621-$782 per tonne in the selected locations, a steep increase from 2020 ($428-$547/t) due to a surge in commodity prices - notably metallurgical coal and iron ore[39]. Whilst there is uncertainty in projected commodity prices and market dynamics, increasing fossil fuel prices and decreasing renewable energy costs will increasingly favour

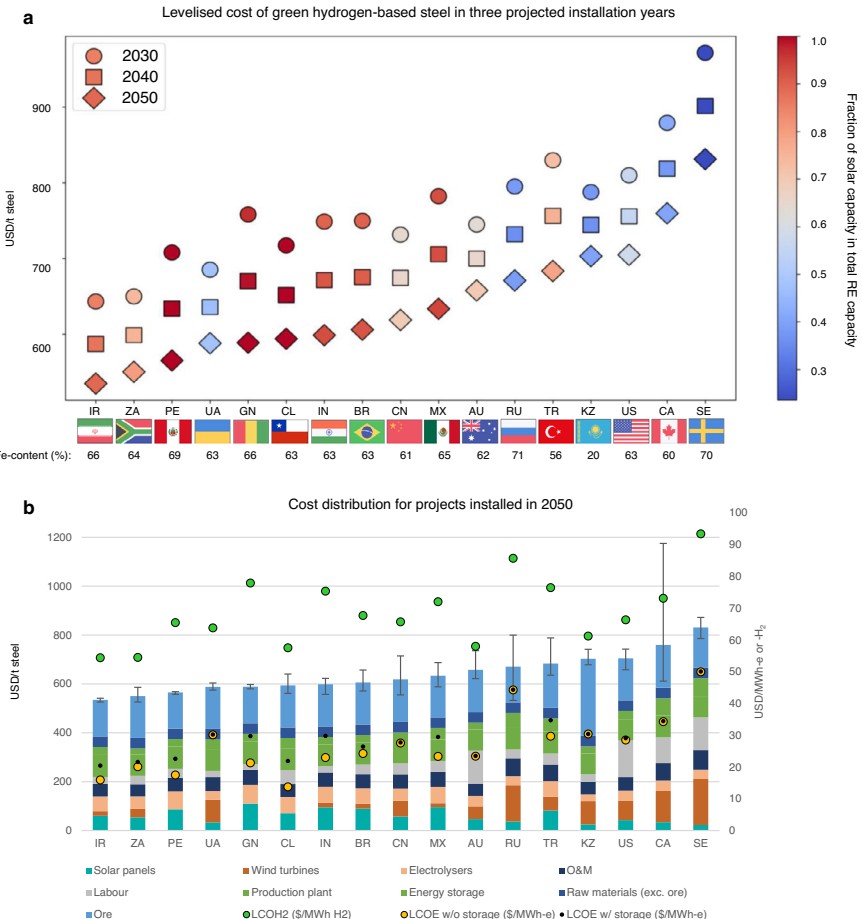

**Fig. 2 | Optimisation results of H₂-DRI-EAF steelmaking with islanded RE supply (0% scrap). a** Cost ranking of 17 countries over three projected years—2030, 2040 and 2050. **b** Decomposition of the LCOS for each country for projects installed in 2050, plotted against the production system's LCOE and LCOH, with error bars showing LCOS interannual and inter-regional variation. Canada has an especially large error bar due to the variance in solar and wind energy across the large land mass where various iron ore deposits are distributed. All costs shown in Fig. 2 (and in all other figures based on optimisation results) were obtained from averaging the results of optimisation runs using five different years of historic RE data (see Experimental Procedure for details), and for a country with multiple iron ore deposit regions, cost figures at the country-level were obtained by averaging the regional results.

the green-hydrogen based steel production route. With reference to the BF-BOF costs at the level of 2021, it was projected that in 2030, a number of studied regions will see market competitiveness (i.e. <$782/t) of green H₂-based steel powered by islanded solar and onshore wind (without scrap charging nor a carbon tax). To reduce the cost premium between fossil-fuelled and renewables-based production, which would be significant if BF-BOF production costs returned to 2020 levels, carbon pricing mechanisms would become essential.

EAF scrap charging can drive cost benefits, so long as contaminants are controlled to avoid affecting steel quality and product lines. Scrap addition generally reduces the LCOS, however, benefits lessen over time with cheaper renewable energy. In 2030, a clear relationship is evident between scrap addition and production costs, a result of the reduced DRI requirement and EAF energy demand (25% scrap addition drives 5% cost reductions, 50% scrap addition drives 9% cost reductions). Towards 2050, however, scrap is less influential in reducing production costs due to the decrease in RE costs which makes green DRI cost-effective and comparable to scrap (50% scrap addition drives 2% cost reduction). Scrap charging into the EAF favoured countries with cheap scrap: China ($212/t scrap), Sweden ($355/t), Brazil ($380/t) and Chile ($387/t), whilst disadvantaging those with expensive scrap: Russia ($624/t), Ukraine ($534/t) and Canada ($472/t). Whilst scrap addition may aid in cost reduction it cannot be relied upon as supply is constrained by historical steel consumption

rates and lifetimes of in-use steel stocks. In addition, BF-BOF production similarly benefits from the inclusion of scrap.

Despite green hydrogen-based steel production generally consuming far less energy than the BF-BOF route (see Fig. S2), the switch in dominant energy source from thermal to electrical requires optimisation of the renewable electricity system and minimisation of supporting infrastructure. Whilst energy represents 8-20% of BF-BOF production costs[39], in the green H₂-DRI-EAF route, the renewable energy system (solar panels, wind turbines, electrolysers) occupies 21-33% of total costs in 2050 (27-41% in 2030). Over the two modelled decades, average costs of projects installed in 2040 and 2050 dropped by 8% and 16% (compared to 2030), respectively, in line with long-term projections of reduced unit costs for RE infrastructure. Accordingly, the average LCOE and LCOH₂ (across all regions) reduced from $43/MWh and $3.2/kg H₂ in 2030 down to $30/MWh and $2.1/kg H₂ in 2050. Although the LCOE and LCOH₂ were determined by the cost-minimising model in this work, it is likely that they will become cheaper given optimal location of solar and wind plants (i.e., not necessarily at the iron ore mine); the global weighted average LCOE of new solar PV and onshore wind projects in 2021 were $48/MWh and $33/MWh, respectively[40].

Green H₂-based steel costs (in 2050, without scrap charging) were accounted to iron ore (28%), solar panels and wind turbines (19%), electrolysers (9%), production plant (14%), energy storage (6%), labour

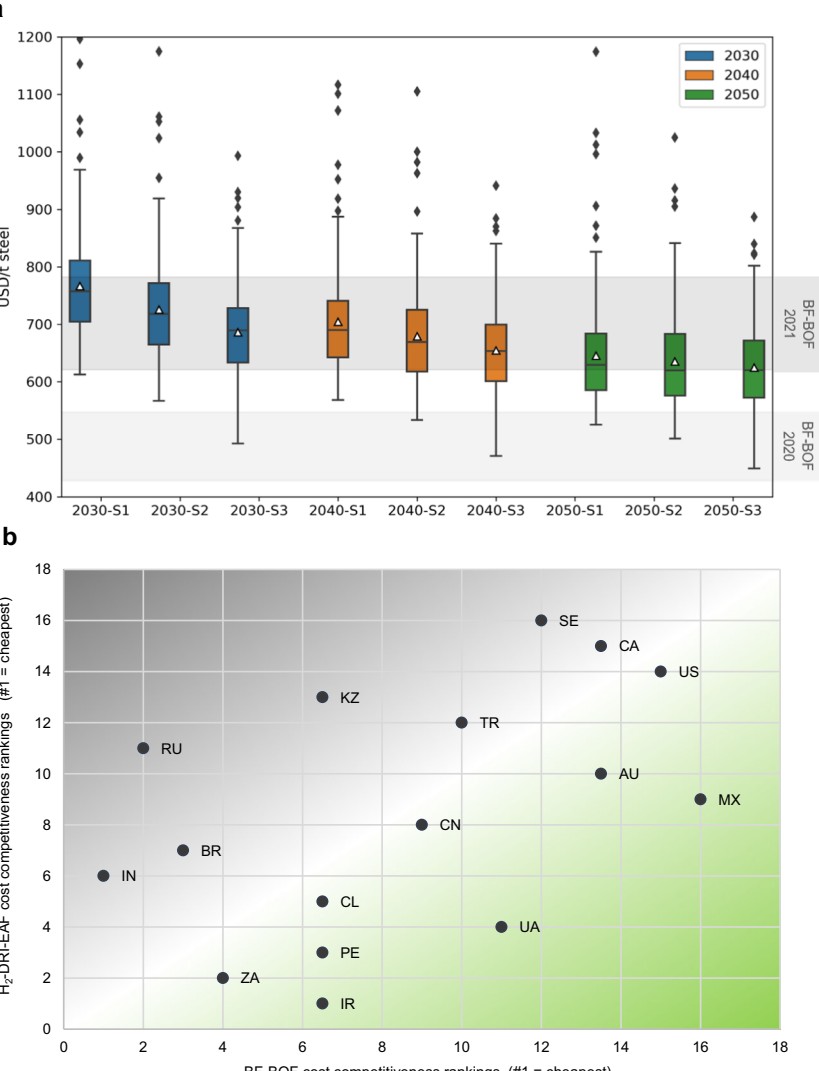

**Fig. 3 | Competitiveness of green H$_2$-based steel with conventional BF-BOF production. a** Steel production costs, at site (inc. ore) (case S1 = 0% scrap, S2 = 25% scrap, S3 = 50% scrap) compared to BF-BOF costs in 2020 and 2021. For each box and whisker plot, $n = 220$ (44 regions each modelled over 5 historical RE data years), and the mean LCOS ranges from $766 to $625. **b** Relative cost competitiveness between BF-BOF (in 2021) and green H$_2$-DRI-EAF (in 2050) production for all 17 optimised countries, excluding Guinea where steel industry currently doesn't exist. Competitiveness ranking is in descending order from least expensive (#1) to most expensive (#16) steel production. BF-BOF cost data from Transition Zero (2022). The closer the country is placed to the bottom right-hand corner, the more it benefits from the BF-BOF to green H$_2$-DRI-EAF transition.

(9%), other raw materials (e.g. alloys, fluxes) (7%) and operation and maintenance (O&M) (9%) (refer to Fig. S1). Ore was by far the largest expense item, with greatest variance between region: 20-45% of the total cost. Significant variance in RE infrastructure cost distribution was projected: 3-19% for solar panels, 0–23% for wind turbines and 4-13% electrolysers. In addition, labour costs were material in differentiating regions; steelworker wages are typically 30% above average[30] and H$_2$, iron and steel production are relatively labour-intensive (refer to Table S19). Labour costs constituted 4-21% of total production costs with high wages disadvantaging Australia, Canada, Sweden and the US (refer to Table S18). If labour costs were excluded, Australia would move up from the 11$^{th}$ to the 3$^{rd}$ lowest-cost producer.

Finally, a transition from the conventional BF-BOF production to green H$_2$-based steel may impact relative regional competitiveness. As illustrated in Fig. 3b, Mexico and Ukraine may benefit most prominently from the transition, followed by Iran, Australia, and Peru. In contrast, Russia, Kazakhstan, India, Brazil, and Sweden are likely to decrease in relative market competitiveness if H$_2$-DRI-EAF steel

production was established using an islanded solar and wind energy system. However, some of these countries may alternatively tap into low-carbon and affordable grid-powered electricity (explored in the following section). Countries like Russia may lose their low-cost steelmaking advantage derived from cheap metallurgical coal and natural gas resources, although, competitiveness may be retained through optimising the use of blue H$_2$ (i.e., H$_2$ production via steam methane reforming with CCS). Green hydrogen-based steel is likely to shift profitable steelmaking dynamics and open new market opportunities; taking advantage of locally available renewable energy resources is critical to maintaining and growing market share.

## Securing cheap, reliable, renewable power for localised production

Affordable, reliable, and wholly renewable electricity is fundamental to competitive near zero-carbon steelmaking. To explore the viability of potential future energy production systems, we compared green H$_2$-DRI-EAF production using both (variable-load) islanded and

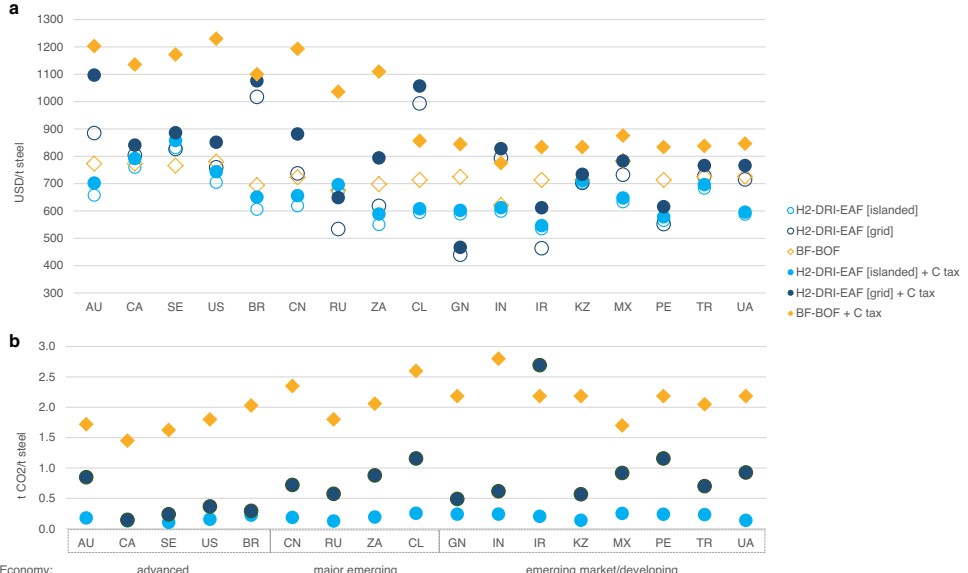

**Fig. 4 | Comparison of conventional fossil-based and novel green H₂-based steel production costs and emissions (projects installed in 2050, no scrap charging).** **a** Projected LCOS, with and without carbon taxes, and **b** CO₂ emission-intensities, where the H₂-DRI-EAF route is modelled with a variable-load islanded energy system, and a continuous-load grid-connected power system. Refer to Figs. S3, S4 for equivalent graphs for 2030 and 2040 project installation years, respectively.

(continuous-load) grid energy systems against the conventional BF-BOF production, as shown in Fig. 4. Unlike the islanded system, energy generation costs of the grid system for 2030 and 2050 are difficult to estimate, therefore the current industrial electricity tariffs were used to indicate the energy cost of the grid system, which vary significantly from $169/MWh in Brazil to $41/MWh in Russia, reflecting the current national energy mixes and polices. They were applied in conjunction with projected grid carbon-intensities, which reflect future energy mixes, and carbon taxes, to investigate the required renewables-dominate grid power prices to support a competitive decarbonised steel industry.

By 2050, it was projected that in 7-of-17 studied iron ore-producing countries, at least one of the two green-H₂ based steel production options (i.e. islanded and grid-based) will become competitive with the BF-BOF option, even without the aid of carbon tax. Six countries were projected to produce green-H₂ based steel using the grid-powered system into 2050 as the cheapest option (Guinea, Iran, Kazakhstan, Peru, Russia, and Sweden), however, when carbon prices were accounted for, grid-powered production remained cheaper in just Russia and Guinea. The generally favourable comparison of the variable-load islanded versus continuous-load grid systems stemmed from two aspects. Firstly, although variable-load islanded systems required additional CAPEX to oversize electrolyser and EAF capacities and provide CGH₂ and battery storage, the cost of RE production was favourable compared to grid power prices. In 2050, the average steel production cost savings under the flexible-load islanded system was $180/t credit for energy, compared to the $100/t debit for increased capacity of flexible processes and energy/material storage. These relationships would change rapidly given a change in industrial electricity pricing; for grid-powered H₂-DRI-EAF production, 10% change in power price causes a 4% change in LCOS. Secondly, the high projected fossil-fuel shares of many national grids led to significant cost penalties by carbon tax (see Fig. 4b). Most prominent was Iran's projected reliance on gas-powered electricity meant that even by 2050, the forecast grid emission-intensity was greater than the BF-BOF route at 2.7 t CO₂/t steel (the average for all other countries in 2050 was 0.7 t CO₂/t steel). In countries where the grid is not decarbonising at a swift pace, long-term renewable energy contracts between steel producers and RE providers, or islanded RE systems, will be necessary to produce near-

zero emissions steel. In contrast, grid-based systems in Canada and Sweden (closely followed by Brazil) may emit just 0.2 t CO₂/t steel in 2030 (see Fig. S3b) due to their grid portfolios with substantial hydropower and nuclear shares. This is already comparable with the islanded system which has emission intensities in the range of 0.1–0.3 t CO₂/t steel across all cases, considering the embodied carbon within solar panels and wind turbines.

The cost competitiveness of the green H₂-DRI-EAF steel production hangs on cheap, CO₂-free power where the variability of renewable resources has been managed. For the islanded system, flexible hydrogen production was far more important than flexible steelmaking to drive down costs; electrolyser oversizing factors ranged from 1.3 to 3.7 (with particularly high oversizing factors for solar-dependent cases, see Table S2) whilst EAFs, following continuous DRI production, were oversized far more modestly from 1.1 to 1.4. Whilst we investigated solar and wind resources, other stable and clean electricity sources (i.e. hydropower) are ideally placed for electricity-intensive steel production. For the grid system, operators must concentrate efforts to ensure energy-intensive industries benefit from cheap RE where variability is effectively balanced across time and space. If a renewables-dominate grid could be secured to power H₂-DRI-EAF steel production, an average global electricity price of $80, $70 and $60/MWh (with the lowest being $62, $54, and $46 $/MWh) would be required in 2030, 2040 and 2050, respectively, to equalise the LCOS (without carbon taxes) across both islanded and grid systems. This is achievable if electricity markets undergo the major anticipated reform in response to growing shares of cheap RE. In 2050, the largest gaps between projected islanded and grid-powered steel costs, and hence between current and target industrial electricity tariffs for competitive grid-powered green H₂-DRI-EAF steelmaking, were observed in Brazil, Chile, and Australia. In these countries, islanded renewable energy systems may be more effective in enabling near-zero carbon manufacturing.

### Global opportunities for green H₂-based steel production
Rapid expansion of LCOS projections from 44 regions to >300 iron ore deposits was achieved using a machine learning (ML) model. The ML model was trained using the optimisation results (ML targets) alongside statistical data of solar and onshore wind potential (ML features).

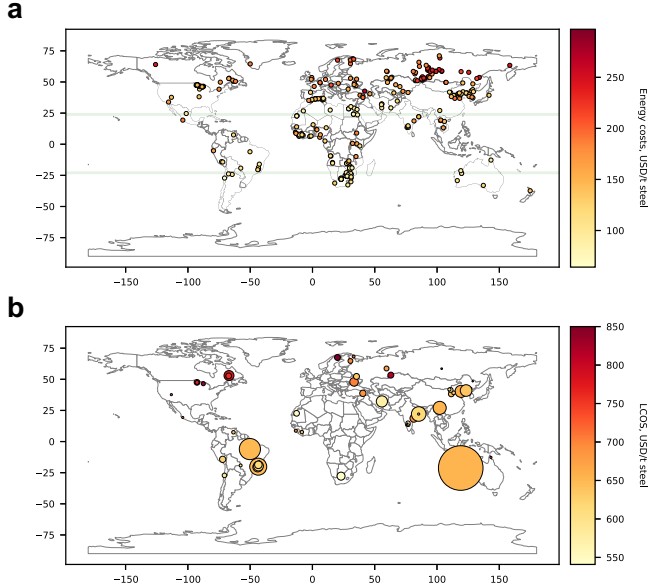

**Fig. 5 | Global ML-projected green H₂-DRI-EAF steel costs using islanded solar and/or wind energy systems (projects installed in 2050, no scrap charging).** **a** Renewable energy infrastructure costs (i.e. solar panels and wind turbines), for > 300 iron ore deposits. **b** LCOS including ore, with markers sized by relative quantity of ore mined on annual basis (mine production data from CRU Group[69] and U.S. Geological Survey[22]). Geographical coverage shrinks from 68 to 22 countries which includes all optimised countries, excluding Guinea. LCOS at Kiruna in Sweden was reduced to $850/t (from $940/t), which is closer to the optimised LCOS, to control the extreme outlier and enable greater colour graduations over remaining mines (the ML model accuracy was reduced in this extreme northerly location).

ML model accuracy was high, demonstrated by a coefficient of variation ($R^2$) value of 0.96 for predicting the levelised cost of renewable energy infrastructure (RE cost) ($8/t standard error, 5% of mean) and 0.85 for predicting LCOS (excluding iron ore and labour costs) ($26/t standard error, 5% of mean) for 1 Mtpa green H₂-DRI-EAF facilities. The cost of solar panels and wind turbines were separated as core cost components requiring further investigation; in 2050, the projected RE costs constituted approximately 20% of optimised LCOS with expected variability (average $120 +/- $35/t steel). Both the RE cost and LCOS ML models may be used to aid future supply chain modelling.

The ML models allowed the development of a global picture of green-H₂ based steel potential (see Fig. 5), revealing that favourable locations were located along the tropic of Capricorn (+23.5° latitude) and Cancer (-23.5° latitude) where strong solar irradiation is readily accessible. Competitive green H₂-based steel clusters were in Northern and Southern Africa, the central region of South America, Central Asia, and Australia. Interestingly, countries dominating current iron ore production appeared to correlate with competitive green H₂-DRI-EAF steel locations, suggesting that for those regions the green manufacturing opportunity is plausible at scale. A striking opportunity exists in Western Australia, a region offering a stable investment environment, so long as beneficiation can manage the progressive degradation of Pilbara ores.

Additionally, the ML model allowed greater understanding of the influence of RE on green H₂-DRI-EAF steel costs. Multicollinearity analysis of statistical RE data determined the following variables to be selected as features (alongside project installation year) based on their degree of independence: (i) mean hourly solar capacity factor (CF), (ii) coefficient of variation (CoV) of monthly solar CF, (iii) CoV of monthly wind CF and (iv) mean monthly wind CF (see Fig. S5), where a CoV indicates the degree of variation in the (monthly) data. Feature

importance analysis of the LCOS model showed that CoV of monthly solar CF was most important for making predictions, followed by installation year, mean monthly wind CF, CoV of monthly wind CF and median hourly solar CF (very similar results for the RE cost model, see Fig. S6). Overall, this analysis confirmed the earlier observation from the results of the optimisation model (Section 2.1): low-variability and plentiful solar resources were more important than wind in establishing a cost-effective green H₂-DRI-EAF steel plant, and wind played more a complementary role in RE generation.

**Production system feasibility at-scale**
Up to this point, our global assessments have been made based on steel production facilities with 1 Mtpa capacity, allowing an 'apples to apples' cost comparison. However, significant growth in green H₂-DRI-EAF steel manufacturing in certain regions could be hindered by resource constraints and industrial development status. To assess the production system feasibility at scale, national green H₂-DRI-EAF steel industries were sized according to the hypothetical utilisation of extracted ore given the following rates of technology diffusion (i.e. H₂-DRI-EAF steel output of total steelmaking potential): 30% in 2030, 50% in 2040 and 60% in 2050. Using our optimisation modelling results (with 25% scrap charge to EAF), an indicative picture of resource requirements is provided in Table 1 for 2050 (with complete analysis given in Supplementary Data). Land intensity rates of 45 MW/km² and 8 MW/km² for solar panels and onshore wind turbines, respectively, were assumed[41], alongside a water demand rate of 12 L/kg H₂ for electrolysis (considering 33% losses and 9 L/kg stoichiometric minimum) and water recycling rate of 9 L/kg H₂ during DRI. Land availability for RE infrastructure was determined within the regions where iron ore mines exist (rather than the entire country) and constrained by 50% of the available shrubland, herbaceous vegetation and sparse vegetation given by the Copernicus Global Land Cover Map[42].

At the assumed level of upscaling, different countries could face a diverse range of resource stresses, such as ore availability, scrap availability, land-intensity (islanded system), grid capacity expansion (grid system) and freshwater demand. While the limitation in the size of ore reserve represents a hard constraint, other stresses could be addressed by measures such as international trade of scrap, significant expansion of power generation capacity, use of alternative low-carbon energy (e.g. nuclear, to reduce land requirement), and use of non-fresh water (and dealing with the additional desalination energy requirement) or implementation of water recycling (to reuse water resulting from iron ore reduction for electrolysis, taking the advantage of co-locating energy and steel production).

In positioning iron ore production as the precursor for green H₂-based steel production, certain countries are confronted with the transition from a raw material export-driven economy to one that is driven by green manufacturing. Australia (and to a reduced extent, Brazil) emerges as a potential future leader in green H₂-based steel manufacturing and exports, given the projection of reasonably competitive green H₂-DRI-EAF production costs and extensive iron ore resources. Fulfilment of vast manufacturing potentials would require significant expansion of renewable energy production. In Australia, this will be most logically supported by islanded energy systems, given that over 4 times the national energy grid capacity would be required in 2050 (assuming 60% technology diffusion), and that iron ore mines are not closely located to established grids. A projected 70-fold increase in Australia's current steel production over the next 3 decades (or even a fraction of this) would need concerted investments and collaboration across private and public sectors, but is far from impossible; the US shale gas industry escalated from near zero output in 2004 to around 2 billion m³ per day in 2020[43]. The greatest challenge facing the country to lock-in favourable economics is to ensure lower-quality ore resources can be directly reduced (e.g. by utilising fluidised bed reactors or additional melting processes to remove impurities)

**Table 1 | Feasibility considerations for green H2-DRI-EAF production using islanded RE systems, at-scale**

| Country | Ore production, 2020 | Hypothetical H2-DRI-EAF steel production | Gross iron in reserves | Iron ore reserve life remaining | Steel production expansion factor (by output) | Scrap surplus | Land consumption of RE infrastructure | Ratio od RE land demand to regional availability | Net water demand | Net water demand as factor of current industrial withdrawal | Energy demand | Ratio of energy demand to current national consumption |
|---|---|---|---|---|---|---|---|---|---|---|---|---|
| | Mtpa | Mtpa | Mt | years | – | Mtpa | 1000 km² | – | 10⁹ m³/yr | | TWh/yr | – |
| Australia | 912 | 395 | 25000 | 44 | 71.05 | -113 | 41.78 | 0.04 | 76 | 27.76 | 1122 | 4.19 |
| Brazil | 388 | 173 | 15000 | 61 | 5.07 | -32 | 15.43 | 0.05 | 38 | 3.64 | 491 | 0.75 |
| Canada | 60 | 25 | 2300 | 64 | 1.98 | -2 | 5.18 | 0.03 | 5 | 0.18 | 72 | 0.11 |
| Chile | 16 | 7 | - | n/a | 5.85 | -1 | 0.33 | 0.01 | 2 | 1.1 | 20 | 0.22 |
| China | 360 | 157 | 6900 | 31 | 0.16 | 431 | 20.49 | 0.17 | 32 | n/a | 447 | 0.05 |
| Guinea | 60 | 28 | - | n/a | 0 | -8 | 2.05 | 0.72 | 7 | 110.79 | 79 | 0.85 |
| India | 204 | 89 | 3400 | 27 | 0.82 | 83 | 7.96 | 0.62 | 21 | 1.21 | 252 | 0.15 |
| Iran | 50 | 23 | 1500 | 46 | 0.88 | -7 | 1.53 | 0.02 | 5 | 4.71 | 65 | 0.18 |
| Kazakhstan | 63 | 9 | 900 | 71 | 2.09 | -3 | 1.34 | 0.02 | 2 | 0.29 | 25 | 0.22 |
| Mexico | 15 | 7 | - | n/a | 0.35 | 11 | 0.57 | 0.16 | 2 | 0.18 | 19 | 0.06 |
| Peru | 13 | 6 | 1500 | 169 | 5.53 | 2 | 0.36 | 0.04 | 1 | 0.43 | 18 | 0.31 |
| Russia | 100 | 49 | 14000 | 201 | 0.67 | 14 | 11.51 | 1.93 | 9 | 0.32 | 138 | 0.12 |
| South Africa | 56 | 25 | 670 | 19 | 4.47 | 15 | 2.12 | 0.01 | 5 | 1.27 | 70 | 0.29 |
| Sweden | 36 | 18 | 600 | 24 | 3.8 | -2 | 5.05 | 0.3 | 4 | 2.62 | 51 | 0.3 |
| Turkey | 15 | 6 | 38 | 4 | 0.16 | 23 | 0.81 | 0.33 | 1 | 1.35 | 17 | 0.05 |
| Ukraine | 79 | 34 | 2300 | 47 | 1.64 | -8 | 5.34 | 1.88 | 7 | 1.55 | 98 | 0.63 |
| United States | 38 | 17 | 1000 | 41 | 0.2 | 83 | 2.37 | 0.22 | 3 | 0.02 | 48 | 0.01 |

Analysis is based on forecasted steel output, determined to be a function of current ore output. Data shown for 2050 production system which assumes utilisation of 60% of national produced iron ore and 25% scrap charge to EAF (refer to Supplementary Data for complete analysis). Negative values in the "scrap surplus" column indicate deficits.

and/or effective ore beneficiation processes are integrated into mainstream mining operations.

Two other countries are also worth mentioning: Sweden, where \$37b of green steel investment has already been directed[16], has very high quality ore reserves but of modest magnitudes, which will limit the scope for producing green $H_2$-based steel from local ore. Also in China, which is currently dominating global steel production (53%) to primarily serve their extensive domestic markets, 90% of its steel production is currently primary (ore-based) production[3]. In the transition to green $H_2$ based steel production, increasing the mix of scrap in the feed to EAF would be beneficial to capitalise on the immense volumes of steel scrap available.

In addition to the resource requirements discussed above, other considerations for the successful transition to green $H_2$-based steel include: (i) reskilling and redistributing steelworkers, (ii) land competition between renewable energy infrastructure, agriculture, and $CO_2$ sequestration[44], and (iii) supply chain bottlenecks including rare earth mineral supply for solar panels, electrolysers and batteries[45]. These factors deserve careful treatments in future business and policy decisions.

## Discussion

The favourable locations identified in this work for co-locating the production of iron ore, RE and steelmaking represent areas where such a simple supply chain is likely to make economic sense. For such cases, product shipping costs still need to be added to the overall costs. Fortunately, ideal locations for green $H_2$-DRI-EAF facilities were within reasonable distance to the coast and its ports. Added costs for FOB (free on board) and CFR (cost and freight to Qingdao, China) were minimal: global average of 3% added cost burden for inland transport and 7% for marine transportation (refer to Table S3). It is unlikely that transport of product will dictate production facility locations, but proximity and regional free trade agreements will support strategic investments. On the other hand, the trade and transport of primary or intermediary products (iron ore, green $H_2$ or HBI) should be explored. Depending on the relative location of high-quality iron ore, renewable energy, and demand markets, steelmaking facilities may not necessarily need to be placed in locations optimal for production efficiency, but rather a favourable location close to critical supply chain links.

In this work, the optimised LCOS using the islanded production system was primarily affected by the regional quality of RE and iron ore, the year of installation and the fraction of scrap. A comparison with literature, where projected LCOS of $H_2$-DRI-EAF steel ranged from \$526[32] to \$778[46], suggests that differences in cost projections are present due to different assumptions with respect to the RE and steel production system and costing basis (see SI Section S2.1 for details). Here, we discuss several factors particularly important for the costing undertaken in this work.

### Iron ore quality at mine-level

Whilst spatial resolution of renewable energy data was at coordinate-level, the best available ore quality data was at national-level, limiting the accuracy of beneficiated ore costs. In addition, characterisation of ore deposits according to composition (haematite, magnetite, goethite, limonite, etc.) was scarce, and consequently the assumption made that all ore reserves were haematite (the most common iron ore type). Given better data, we would have performed differential beneficiation and reduction analysis to determine more accurate production costs. The simplified linear cost equations for DR-grade ore applied in this study are illustrative of the added cost burden of beneficiation, but not conclusive.

### RE potential

The nature of optimisation modelling using historical RE data caused the assumption of planned flexibility, which is impractical. Inter-annual

variability was addressed in this work by running the optimisation models repeatedly with different historic annual RE profiles across five years (2015-2019); the difference in LCOS projections between repeated runs appear to be modest across most locations (see error bars in Fig. 2b). Nevertheless, modelling over a longer period, e.g. 20 years of historical RE data could yield results that can be compared with those from this work to further increase confidence. Furthermore, this global study considered solar and (onshore) wind as RE sources for the islanded system; further detailed regional studies could explore other renewable energy resource integration, namely offshore wind, hydropower, and biomass.

### Industrial electricity tariffs

For assessing grid-based systems, we used current industrial power prices (of 2018), in absence of clear projections of future power prices, to reflect on the energy system conditions for competitive steelmaking, and hence required electricity market reform. These tariffs are highly unlikely to stay constant due to changes in energy portfolios, commodity prices (namely natural gas and coal), network costs (likely enhanced distribution infrastructure due to integrated RE systems), need for RE storage, and taxes/levies. National grids must deal with transitional challenges including RE supply scarcity aligning with peak demand. Steelmaking facilities may be able to take advantage of dynamic pricing, where production is aligned with demand troughs, or establish long-term renewable energy contracts with energy providers to receive stable, low-cost renewable electricity.

### Flexibility in green $H_2$-based steel production

This work considered flexible operation of the electrolyser and the EAF. Other means of flexibility left unexplored include altering the load factor of the DRI shaft furnace (e.g. 70–100%) and allowing a flexible scrap charge for every batch (scrap fraction becomes an annual rather than hourly constraint, allowing the hourly level to vary). Ideally, scrap charge to the EAF should be set based on regional scrap flows, including import and export. Whilst we explore islanded and grid-powered energy systems separately, semi-islanded energy systems with varying degrees of dispatchable power may lead to cost reductions.

Finally, several further challenges of $H_2$-DRI-EAF steel making require attention. In this work, we have considered the beneficiation requirements to uplift the quality of the currently available ore to DR-grade. Considering that DR-grade ore constitutes just 4% of current global seaborne trade, and accessible high-grade ore reserves are decreasing with cumulative extraction[47], scale-up of iron ore beneficiation competence and capacity is a global priority. Especially in Australian Pilbara ores (responsible for 37% of annual global iron ore output[22]), the increasing presence of goethite (an iron oxyhydroxide) in extracted ore is a concern due to its high porosity and friable texture, leading to ultrafines generation[48]. Whilst ultrafines improve sinter production efficiency, they must be avoided in DR shaft furnace operation due to the counterflow reduction mechanism. Iron ore producers are aware of the threat of increasing DR production to their product marketability; ore reserves suitable to serve the DR-grade market remain limited[49].

If the raw material challenge cannot be met by supply, downstream processing must become more flexible. Two flexible processing options have emerged: (i) BF-grade iron ore pellets may be used in the DR shaft furnace, however, with an additional process pre-EAF to melt the sponge iron and remove impurities via slag formation (DRI-melter-EAF), and (ii) the use of DR-grade (or potentially BF-grade) iron ore fines in DR with fluidised bed reactors, so long as the fluidisation velocity is maintained above the minimum and the agglomeration phenomenon of 'sticking' (increased adhesion and friction among particles) is controlled[49,50]. Nevertheless, both supply- and demand-side

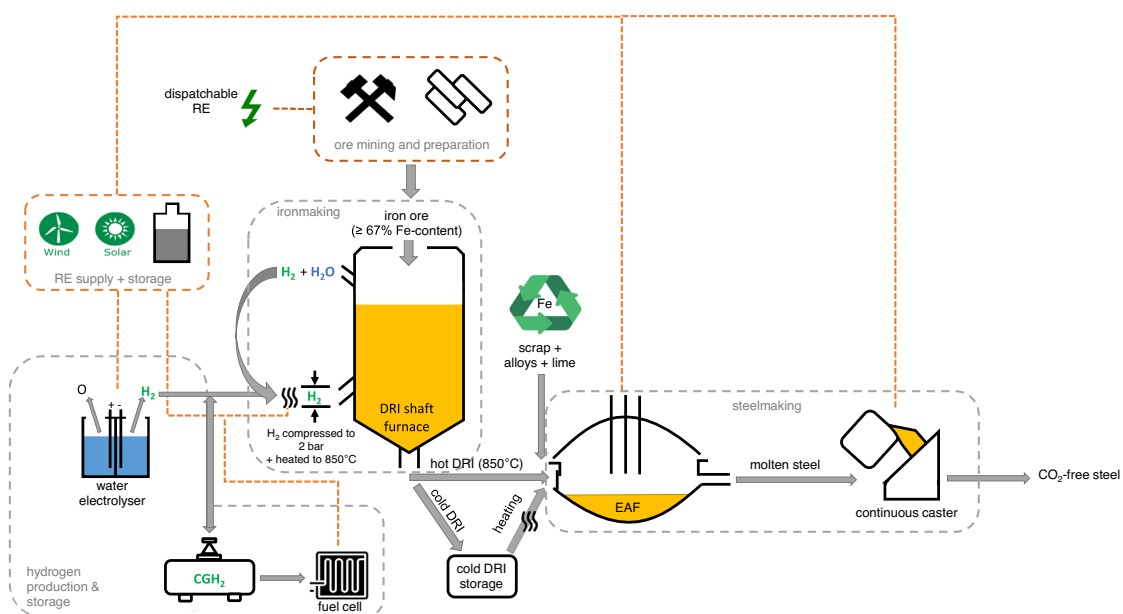

**Fig. 6 | The modelled *integrated* green H$_2$-*DRI-EAF* steel production system.** The overall production system consisted of five sub-systems: RE supply and battery storage, hydrogen production and storage, *iron* ore mining and preparation, ironmaking, and steelmaking. All processes are co-located and the RE supply is fed to all other sub-systems which are further discussed below (also refer to Table S4).

innovations are required to enable productive green H$_2$-based DRI-EAF supply chains.

In addition to the critical issues around iron ore supply, several other challenges have been identified for the green H$_2$-DRI-EAF route, most notably the maintenance of metallisation degree in the shaft furnace in light of the resistance of diffusion of H$_2$, and the different melting characteristics of carbon-lean H$_2$-DRI in the EAF[15]. Despite the current success of H$_2$-DRI-EAF with polit-scale testing as demonstrated in HYBRIT[51], further technical developments are still required to fully resolve these issues.

## Methods
### Description of the steelmaking system
Green H$_2$-DRI-EAF steel production with various shares of scrap (0%, 25% and 50%) were modelled, alongside 2030, 2040 and 2050 project installation years. The co-located supply chain spanned from iron ore mining and renewable energy generation to the production of semi-finished steel products (i.e., slabs, billets, and blooms) (see Fig. 6).

**RE supply and battery storage.** Solar and wind resources directly energised electrical processes, and indirectly energised chemical processes via green hydrogen production. Storage of electrical energy was enabled through lithium-ion battery integration (85% charge/discharge cycle efficiency) or compressed gaseous hydrogen (CGH$_2$) converted via the fuel cell to electrical energy (~40% round-trip efficiency). A lithium ion battery was selected for electricity storage due to its relative high efficiency, prolonged cycle life (up to 10,000 h at 100% depth of discharge) and intermediate self-discharge rate (5–8% per month at 21°C) in comparison to other battery technology[52, 53].

**Hydrogen production and storage.** Green hydrogen is produced through renewable-powered water electrolysis. Multiple electrolysers are in development with key differences in the current/projected energy efficiency, unit cost, cold start time and lifetime. Low-temperature (80°C) alkaline technology is mature with reasonable dispatchability to respond to variable renewable energy inputs[54], and hence was selected for this study. Due to electrolyser modularity, electrolyser capacity was treated as a continuous variable. CGH$_2$ stored

aboveground in steel tanks at 200 bar pressure was incorporated into the energy system as both a H$_2$-buffer (pre-DRI) and energy store when combined with fuel cell stacks. Although underground CGH$_2$ storage may be preferred on an energy-efficiency basis for seasonal storage (lower pressures required), a storage option with rapid charge/discharge rates was prioritised (storing H$_2$ in geological formations, an option that may become relevant for large-scale implementation, is discussed in SI Section S2.2).

**Iron ore mining and preparation.** Iron ore is a naturally occurring geological resource from which metallic iron (Fe) can be extracted, most commonly present as haematite (Fe$_2$O$_3$) or magnetite (Fe$_3$O$_4$) with varying levels of Fe-content (global average of 62%)[22]. The ore is then processed to produce a marketable product of specific physical form (lump, pellet, fines) and Fe content (DR-grade ore ≥67% Fe). Beneficiation, a process whereby gangue material is removed and Fe concentration increased, is proceeded by pelletisation to produce DR-grade ore. Depending on the degree of beneficiation required, crushing, screening, and grinding processes are utilised in conjunction with gravity or magnetic separation. Whilst very high quality deposits can be mined and used as lump, DRI shaft furnaces are technically constrained to a lump-to-pellet ratio of 3:7 to prevent ore clustering[21].

**Ironmaking.** DR-grade iron ore is reduced by H$_2$ at a minimum consumption rate of 50 kg H$_2$/t DRI and metallisation rate of 94%[55]. Prior to injection into the shaft furnace, H$_2$ is heated to 900°C (gas acts as reductant and heat carrier) and compressed to 2 bar (to overcome furnace pressure drop). DRI operations are generally continuous with minimum load variation to ensure a uniform oxidation front. The outputted sponge iron can be fed directly to the EAF (hot-DRI at 850°C), stored on site (cold-DRI) for later re-heating and EAF charging, or modified into a more stable product as hot briquetted iron (HBI)[56]. Since iron and steelmaking are co-located in this study, only hot-DRI and cold-DRI are utilised. Hydrogen direct reduction processing was based on Vogl, et al.[46], whilst the energy and H$_2$ requirements were determined using Aspen Plus simulation[32].

**Steelmaking.** EAFs are charged with DRI, scrap, alloys, and lime, and consume carbon electrodes to produce molten steel at 1600°C. Metal yield is inhibited by DRI gangue content; impurities can be removed via EAF slag which is stimulated by slag formers such as lime fluxes, although this is an energy-intensive process and impurities are best dealt with during DRI. Increasing the scrap charge will reduce slag demand, lime demand and EAF energy consumption[57]. The EAF operates in batch mode and often a single steel plant will have multiple EAFs of various capacities. The model considered a 60-minute tap-to-tap time for each batch; for simplicity, EAF operation was treated as a continuous variable.

## Location selection and geospatial characterisation

The top-16 iron ore producing countries were selected for analysis, constituting 98% of ore production (by output quantity)[22], with the addition of Guinea, a nation with high quality ore (65.5% Fe) which has not yet been extracted[58]. A country was split into multiple regions when the national deposits were dissimilarly located (classified as when the standard deviation of latitudes and longitudes exceeded 1°), totalling 44 regions (see Table S5).

Solar and onshore wind potential was determined using Renewables Ninja, which provided data for each of the 8760 h across the five selected historical years: 2015-2019. RE data was extracted at coordinate-level for each ore deposit, then regionally aggregated across multiple deposits. Despite publicly available records of the exact coordinates of each iron ore deposit, ore production and reserve data was only available at national level[22]. The applied Fe content was determined by current extracted ore characteristics (as opposed to the reserve characteristics) as this more accurately reflected the economic resource and mine operation feasibility (refer to Table S6 for key geospatial dataset details).

For simplicity, ore cost adjustments were modelled as a function of Fe-content. DR-grade lump ore is a premium product due to the very high reserve grade requirements (≥66% Fe), and DR-grade pellets due to the additional energy (communition, concentration and pelletising) and material inputs (binder). Based on top-down accounting from the 62% Fe market reference price of $100/dry metric tonne (dmt) (10-year average, World Bank Group[59]), ore production costs were assumed to be $60/dmt for DR-grade lump, and an average $100/dmt for DR-grade pellets (Eq. 1). An extra $10/dmt was added to Sweden's site costs due to underground operations and consequent drilling and blasting requirements at Kiruna, the dominant national mine. Ore preparation energy demand was calculated alongside that of iron and steel production (see Tables S7, S8 for mass loss rates and process specific energy consumptions).

Equation 1: DR-grade pellet cost ($/dmt)

$$C_{DRpellet} = 60 + 40 * \frac{Fe_{global}}{Fe_{national}} \quad (1)$$

$Fe_{global}$ = average global Fe content, 62%

$Fe_{national}$ = Fe content of extracted ore(t)/total extracted ore (t)

Apart from these geospatial features, scrap prices and steelworker wages were regionally differentiated (refer to Tables S17, S18, respectively). Scrap steel unit prices were regional variables based on the export price, excluding Turkey (dominant global scrap steel importer) for which import price was used (HS 720429 Waste or scrap, of alloy steel, other than stainless from UNComtrade[60]).

## Optimisation of islanded RE-driven steel making facilities

A linear programming optimisation model of the 1 Mtpa green $H_2$-based steel production facility was developed and solved using GAMS (General Algebraic Modelling System). The model pertained hourly resolution over a characteristic year and was optimised for lowest cost using the CPLEX solver. The objective function (Eq. 2) was to minimise

the steel plant's annualised costs (Cost$_{ann}$), covering both capital expenditure (CAPEX) and operational expenditure (OPEX), given resource consumption parameters (see Table S10) and cost parameters (see Table S11) of the steel production process. The ore subsystem was an externally managed parameter; the quantity of ore required was a function of the steel scrap fraction and extracted ore Fe content. A Net Present Value economic assessment (Eqs. 3–9) was conducted over the 20-year project lifetime given a discount rate of 8%.

For all 8760 hourly timesteps across a year, the model determined the flow of energy to production, storage, or curtailment, which defined overall process capacities and production costs. Each model was run for a given location and RE input year (to consider inter-annual variability) over 9 loops for all combinations of project installation year (2030, 2040, 2050) and scrap input (0%, 25%, 50%). Over the two modelled decades, maturing RE technology (solar panels, wind turbines, li-ion batteries, electrolysers, and fuel cells) benefitted from cost reductions and efficiency improvements (refer to Table S9). Although up to 100% scrap charge can be enabled in the EAF, 50% was chosen as the maximum charge in alignment with the maximum forecasted portion of secondary steel production towards 2050[5]. The principal case for analysis was without scrap since it provided a direct comparison to the conventional BF-BOF route and removed instable scrap price influences. All costs are presented in USD, 2020 dollars.

The decision variables surrounded scheduling and capacity planning of each production unit, more specifically: (i) energy infrastructure capacity (solar panels (MW), wind turbines (MW)), (ii) production infrastructure capacity for flexible processes (electrolysers (MW), $CGH_2$ storage (t), battery storage (MWh), EAF (t), continuous caster (t)), (iii) electricity storage medium (li-ion batteries (MW of 4-hr batteries) or $CGH_2$ storage (t) combined with fuel cells (MW)), and (iv) hot- or cold-DRI feed (t) to the EAF. The primary constraints were 1 Mtpa steel production and operation using a completely islanded energy system (zero dispatchable power integration). Water and land constraints, as well as carbon-intensity, were considered as systems enablers/disablers but not accounted for in the NPV analysis (water accounts for about 1% of total LCOS[32]). Whilst LCOS at site was the primary economic metric, transport costs to the nearest port were included for free on-board (FOB) costs and then marine transportation to Qingdao, China, for combined cost and freight (CFR) to the largest demand centre (see Table S12 for freight rates). Marine transportation was assumed to be powered by green ammonia and solid oxide fuel cell combined with an electric motor.

Equation 2: Objective function

$$Minimise : Cost_{ann} = CAPEX_{ann} + OPEX_{ann} \quad (2)$$

Equation 3: Annualised CAPEX

$$CAPEX_{ann} = f^{CR} * (CAPEX_{production} + CAPEX_{storage} + CAPEX_{energy}) \quad (3)$$

Equation 4: Capital recovery factor

$$f^{CR} = \frac{r(1+r)^n}{(1+r)^n - 1} \quad (4)$$

Where $f^{CR}$ denotes the capital recovery factor, $r$ represents the real discount rate (8%) and $n$ denotes the project lifetime (20 years).

Equation 5: CAPEX of production and storage infrastructure

$$CAPEX_{production} = \sum_i (capacity_i * unit\ cost_i) \quad (5)$$

Where $i$ denotes electrolysers, $H_2$ heater, DR shaft furnace, EAF and continuous caster.

Equation 6: CAPEX of storage infrastructure

$$CAPEX_{storage} = \sum_j (capacity_j * unit\ cost_j) \qquad (6)$$

Where $j$ denotes compressor (200 bar), $CGH_2$ storage vessel, fuel cells, and batteries.

Equation 7: CAPEX of energy infrastructure

$$CAPEX_{energy} = (capacity_{solar} * unit\ cost_{solar}) \\ + (capacity_{wind} * unit\ cost_{wind}) \qquad (7)$$

Equation 8: Annualised OPEX

$$OPEX_{ann} = \sum_k (consumption_k * unit\ cost_k) + O_{mnt} \qquad (8)$$

Where $k$ denotes consumables (iron ore, scrap steel, lime, alloys, graphite electrodes) and labour.

Equation 9: Maintenance OPEX

$$O_{mnt} = 0.02 * (CAPEX_{energy} + CAPEX_{production}) \qquad (9)$$

## Modelling the grid-based system and BF-BOF route

To investigate the relative cost competitiveness of various renewable energy systems, "variable-load islanded" was compared to "continuous-load grid" for 1 Mtpa green $H_2$-DRI-EAF facilities (without scrap charging). For the grid-based system, steel costs were differentiated at country-level by current industrial electricity tariffs (ref. 61, refer to Table S13) and projected grid carbon intensities[62]. For islanded green $H_2$-DRI-EAF production, $CO_2$ emissions were also calculated and carbon prices applied, however only for the renewable energy system. 40 g and 10 g $CO_2$/kWh for utility-scale solar panels[63] and wind turbines[64], respectively, were assumed. GHG emissions from electrolysers, and iron and steel production facilities were not considered, as they were in the grid-powered production systems, due to negligible contributions. Projected carbon taxes were applied according to the IEA[65] net zero scenario and economy classification: advanced (Australia, Canada, Sweden, and the US), major emerging economies (Brazil, China, Russia, South Africa) and developing economies (Chile, Guinea, India, Iran, Kazakhstan, Mexico, Peru, Turkey, Ukraine), equal to $250, $200 and $55/t $CO_2$ in 2050, respectively (refer to Table S16).

Both $H_2$-DRI-EAF systems were compared to the conventional BF-BOF route, with unique production costs for each country based on data from Transition Zero[39] (see Table S15). National carbon-intensities of BF-BOF production were given by Hasanbeigi and Springer[66], with economy-wide averages (weighted according to steel output) utilised for unspecified countries: 1.7, 2.3 and 2.6 t $CO_2$/t steel for advanced, emerging and developing, respectively (see Table S14). The marginal abatement costs were also calculated for both decarbonised steel routes, that is the minimum carbon price required to equalise BF-BOF costs.

It is necessary to acknowledge that once the energy system is decarbonised, some minor process emissions will remain within the green $H_2$-DRI-EAF route due to lime (28 kg $CO_2$/t steel), graphite electrodes (6 kg $CO_2$/t steel) and fossil-based carbon injection into EAF (17 kg $CO_2$/t steel)[67]. The decarbonised production route that removes carbon from the direct reduction of iron, however, also removes the necessary carbon source in the EAF to produce the C-Fe alloy that is steel (depending on the steel quality, C content in steel ranges from <0.3–1.5%). In absence of C-containing DRI, carbon can be injected into the EAF. Biogenic carbon (biochar, produced via torrefaction or gasification) can be used instead of coke at a rate of 12 kg/t steel (compared to 8 kg/t steel for coke), procured at a price dependent on the biomass

cost but may be approximated at 235 $/t, equal to less than $3/t steel[67]. If biochar is used in place of coke for essential steel alloying, total process emissions can be reduced to 23 kg $CO_2$/t steel (1% of the BF-BOF route's emissions).

## Building the machine learning model

ML was utilised to develop a rapid green $H_2$-based steel assessment tool, enabling cost projections for >300 locations (covering 68 countries) in less than a second. This was a significant timesaving considering the GAMS optimisation model's computational processing time of 3 h (on a machine with Intel i7-8665U CPU and 16 GB memory of RAM running Windows 10) for a given location and RE input year. Gradient-boosted regression models from the scikit-learn toolkit[68] were fitted to directly predict two targets: levelised cost of renewable energy infrastructure (RE cost), and steel (LCOS, excluding ore and labour) for green $H_2$-DRI-EAF steel production without scrap charging in a 1 Mtpa facility. The ML algorithm learned from a dataset with 675 entries: 45 regions modelled over 5 renewable energy input data years and 3 installation years. Note that New Zealand was the 45th region added to the 44 previously optimised regions to ensure the largest range of latitudes were covered in the input dataset. To determine the overall LCOS, statistical RE data from 2019 was used as features to project the machine-learned LCOS (excluding ore and labour), with separately computed costs of DR-grade ore (see Eq. 1) and labour added.

Regression analysis accuracy relied on detailed characterisation of the renewable energy availability and intermittency at a specific location (becoming the features of the ML model). Renewable energy statistics were computed, encompassing the mean, median, and coefficient of variation of the hourly and monthly capacity factors, for both solar and wind, providing 12 possible features for the ML model to utilise to predict the LCOS and RE cost (aggregated RE statistics for the 17 optimised countries are shown in Table S1). A multicollinearity analysis was undertaken using hierarchical clustering to remove redundant predictors, thereby preventing issues when interpreting feature importance.

A gradient boosting regressor was selected for ML as an additive model (i.e. sum of multiple simple models) that uses decision trees to accurately predict continuous values. Nested cross-validation (CV) was applied to estimate the generalisation performance ($R^2$ value) of the model and select optimised hyperparameters (which control the learning process). Since the regressor's purpose was to model novel locations, the key metric to maximise was the performance of regions not included in the training set. Nested CV involved a series of train/validation/test set splits where data was split into 17 folds such that each fold contained data from a single country. A set of countries were then selected as suitable for testing which excluded extreme regions that we did not expect the model to generalise to. For hyperparameter selection, we searched exhaustively over the Cartesian product of: (i) the learning rate, selected from [0.01, 0.1, 0.5], (ii) the maximum depth of the tree, selected from [3, 5, 10], and (iii) the number of estimators, selected from [100, 5000, 10000]. To obtain the final model, we set the hyperparameters to those that were selected most often in the inner CV and trained the regressor on the entire data set.

## Data availability

All data generated or analysed during this study are included in this published article (and its supplementary information files). Source data are provided with this paper.

## Code availability

Code developed in this study is available online for the optimisation component (https://figshare.com/s/92ed30035b61fd174d93) and machine learning component (https://figshare.com/s/a3849465ee2e09744876).

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

## Acknowledgements

Financial support was provided by the General Sir John Monash Foundation for Alexandra Devlin (for the duration of the PhD studies).

## Author contributions

A.D. conceptualization, methodology, software (GAMS optimisation and machine learning models), analysis, writing—original draft. J.K. software (machine learning model). H.G.-J. investigation (RE data collection & variability impact analysis). A.Y. conceptualization, supervision, validation, writing—review and editing.

## Competing interests

The authors declare no competing interests.
