## [Peer Review File · Nature Communications]

REVIEWER COMMENTS

Reviewer #2 (Remarks to the Author):

Thank you for the invitation to review this interesting and topical paper. I have read it three times and struggle with my assessment. On the one hand it presents really interesting results. On the other hand there are many weaknesses and mistakes. I therefore recommend major revisions. I would really encourage the authors to make an effort to revise so that this can be published soon.

An overarching comment is that the authors should add some caveats concerning uncertainties in the assumptions and discuss the results against those uncertainties. I think the results are pretty robust but things could shift around a little with different assumptions and input data. Another overarching comment is that it is not clear what the machine learning approach actually does or contributes (it is described in language that most readers cannot follow). I would also recommend that the authors go through the text checking that all information given is relevant and important as to me there seems to be unnecessary detail and information.

More specific comments:

Line 11 and 34: You may wish to consult the latest IPCC report for a number on share of emissions

Line 19: The assumptions/discussions around islanded versus grid connected are problematic, see below.

Line 31: present electricity demand also as TWh

Line 39: Avoid using the term “hard to abate” since it is misleading. Is steel harder than anything else? It seems pretty easy through the HDRI-route

Line 41-42: Many would disagree that materials efficiency measures have already been adopted. I think steel is heavily overused and would look for references by Julian Allwood and Jonathan Cullen on this. Among options I think many steelmakers are also considering smelt reduction with BOF to handle lower grade ores. Better check this option.

Line 69: Since beneficiation is so important please consider adding some explanation/detail on this process.

Line 75 onwards: You should take more care in finding and referencing earlier work on this topic. See for example Gielen et al <https://doi.org/10.1111/jiec.12997> and Trollip et al <https://doi.org/10.1080/14693062.2021.2024123> . A proper search may reveal more.

Line 104: What does it add do different levels of scrap? Is this important to keep?

Line 116-117: Even grid connected mills can have 100 % renewable electricity (even if the total grid has emissions) and the price can be set through power purchase agreements rather than follow the tariff or the spot price if there's a power exchange. The distinction made between islanded and grid connected to me is very misleading. This is more about the details and peculiarities around electricity market regulation and design in different countries. You could keep it roughly as it is but make very clear to the reader that you make a very "artificial" assumption around grid connected electricity.

Figure 2b: If I am doing the math right this figure implies LCOH at 2-3 USD/kg in 2050. To me this is much too high when many in the business have 1.5 USD/kg as a target. LCOE at 50 USD in 2050 for Sweden is very high. New wind in northern Sweden today is 25-30 USD/ MWh.

Line 172: \$4280 is a typo?

Line 173-175: What is the relevance of general coal prices? The authors should look at the prices of coking coal (important for the economics of BF-BOF)

Line 197-199: I question the importance or relevance of including scrap shares in the analysis, and in any case scrap prices may change a lot over time.

Line 210-212: LCOE is a key assumption that is worth discussing more. It is quite likely that LCOE (and LCOH) will much lower than assumed here.

Line 219-223: What is the source of labour costs and are the wages representative of skilled steelworkers?

Line 256-263: I think these results should be presented as indicative and with caveats. For example, blue hydrogen may be an option for Russia.

Line 264-265: I think this whole section is misleading and it is mixing cost (in islanded systems) with price in grid systems. Even with grid connection you can have 100 % wind and PV through PPAs. This needs to be rewritten or at least highlight that the distinction made is quite artificial.

Line 332-333: This section can do a better job of explaining what the ML does.

Line 393: Do you consider that water is or can be recycled after the DR-step?

Line 398-405: Why is size of ore reserve a hard constraint when ore is easily traded? What will be the LCOE of nuclear? (BTW be careful with language and distinguish between steel export and HBI export)

Line 410: “expansive” should be “extensive”?

Line 423-424: What does it mean that investment has been “poured”?

Line 442: Ore can also be traded/exported?

Line 492-493: Here you make an important observation that basically undermines much of you discussion/results on grid connected production.

Line 507: of annual global what?

Line 530: There’s a paper by Vogl on this: <https://doi.org/10.1016/j.joule.2021.09.007>

Line 549-560: How much electricity storage at what cost is used in islanded systems? I imagine an islanded system with lots of H2 storage to operate 24/7 and some electricity storage to run pumps, motors, etc. but what roughly is the share of each?

581-588: I wonder if this level of detail is needed and relevant for the aims of the paper.

Line 592: word missing?

Line 685: As noted already current industrial electricity tariffs are not very relevant. Also, it is very risky to mix different sources as in S13.

Line 715: This section is rather incomprehensible to me. Would it be possible to explain the ML in more layman's terms?

I have not scrutinized the supplementary material but note that some tables are missing sources/references.

Reviewer #3 (Remarks to the Author):

Summary

- This is a good paper worthy of publication in Nature Communications after addressing the referees' comments.

Editorial

- Line 67 – “Distinct from” instead of “distinct to”?
- Line 172 Typo on BFBOF steel costs, \$4280-\$547/t
- Line 592 “...NOT yet been extracted” ..
- Line 606 Explain “dmt”.
- Line 706 “THAT removes ...”, or “while removing carbon from direct reduction of iron”

Substantive

- Line 35 – Bataille et al 2021 had three demand forecasts: low (1.9 Gt/yr in 2050, evolving to 200 kg/cap/yr in 2080; 2.2 Gt/Yr 250”; and 2.5 Gt/yr 300”). You are likely referring to the higher value.

- Line 40 – You may wish to distinguish iron ore reduction from steel making here, but perhaps you do this later.
- Line 70 – It may or may not be useful for your paper that Bataille et al 2021 geographically structured its scenarios based on $\leq 100, 200$ and 300 km to known feasible CCS reservoirs, and allocation of HDRI stock to a minimum of 3.5 watts per m^2 per day solar PV using global databases and the locations of existing facilities or a 1-0 national switch denoting access to inexpensive hydropower.
- Line 69-71, Line 159-164 (Kazakhstan) I will be looking to see how beneficiation/ore upgrading is treated
- Lines 220-223 I find it hard to believe wages in Canada ($\$43/hr$) are more than Sweden ($\$35/hr$). You may wish to check this, and how you accounted for relative income tax costs.
- Line 358-375 (Figure B), also Line 473-477. Given the progressive degradation of the Pilbara ores, you should say something about what the huge dark green dot in Western Australia, seemingly the size of the rest put together. Is it beneficiated magnetite, goethite, a mix ?
- Line 434 What is meant by water “abstractions”? Total use? Net loss after recycling? Subtractions?
- Line 434 Gross iron in ore reserves (over a certain concentration?) may be a useful value
- In line 434 I would think carefully about what to add and what to subtract.
- Line 451-464 - Correct me if I am wrong, but all your analysis focuses on trade route where the full H-DRI-EAF process is done in one place, not where sponge or HBI is shipped to EAFs in the target market e.g., (Bataille et al., 2021; Trollip et al., 2022). There are tremendous number of different final steel types and chemical mixes, whereas reduced iron is moved and used in only a few ways, arguably making making shipping of reduced iron for final processing in a BOF or EAF easier for final industrial structure to absorb.
- Line 473 – This is where you will receive your most reader push back, as there is no discussion of differential beneficiation, which differs tremendously by ore type. Arguably, this is the weakest part of the paper, but you do seem to at least acknowledge this from line 500 onward.
- Line 487-493. Industrial facilities DO NOT want to align production to minute by minute, hourly RE availability – that is simply not bankable/investable. However, they can size energy resources and storage to their 24/7 needs, and build that into the business plan (Trollip et al., 2022). For example, Hybrit is building a cavern to store several days of necessary hydrogen, and is planning to arbitrage North Sea wind excesses/low price periods.
- Lines 494-499 I encourage you to investigate this further in future papers.
- Line 525 – Do you have a reference for the different melting points of reduced iron by carbon content?
- Figure 6 Have you included a valuation of the separated oxygen?
- Lines 635-637. The assumption of maximum 50% scrap may hold globally or nationally, but not on a facility basis and is not justified. It should have been set based on regional availability – scrap availability is heavily dependent on level and stage of development, and the possibility of import.

- Line 664 You use an 8% discount rate in the CRF here, and 7% previously (Line 628)? Is this a typo?
- Line 691-694 it might be useful to include the IEA 2021 carbon prices here for reference.

Thank you for a pleasant and informative read.

Bataille, C., Nilsson, L. J., & Jotzo, F. (2021). Industry in a net-zero emissions world: New mitigation pathways, new supply chains, modelling needs and policy implications. *Energy and Climate Change*, 2, 100059. <https://doi.org/10.1016/j.egycc.2021.100059>

Trollip, H., McCall, B., & Bataille, C. (2022). How green primary iron production in South Africa could help global decarbonization. *Climate Policy*, 22(2), 236–247. <https://doi.org/10.1080/14693062.2021.2024123>

Reviewer #4 (Remarks to the Author):

This is a well-written and interesting paper on an important piece of research. I have a number of minor comments that could be addressed in the paper and hopefully strengthen it.

1. A linear relationship is used to calculate DR pellet cost calculation based on iron content. To what extent is this a good fit to observed cost data?
2. In Fig. 2b, the cost uncertainty for Kazakhstan is considerably greater than the other countries, and should possibly be addressed further in the text.
3. Generally the work assumes that green steel will be produced using wind and solar power, and little mention is given to hydropower and nuclear power. It may be worthwhile addressing how the results might change if these other sources are considered. This could be particularly interesting for Sweden, which appears to have the highest LCOE of the 17 countries shown in Fig. 2b but which currently meets the vast majority of its electricity demand using hydro and nuclear.
4. It may be worth briefly mentioning energy storage costs in the results following Fig. 2, as they are currently not shown in the cost breakdown of Fig. 2b or mentioned in the text following the figure.
5. On line 138, I would suggest that “levelised costs of green steel (LCOS)” might make more sense than “green levelised costs of steel (LCOS)”.
6. On line 146, the costs from the IRENA report should be per kW, not per MW.

7. I believe there is a rogue digit in one of the costs on line 172 (currently says \$4280-\$547).
8. In or around the paragraph starting on line 191, it may be worth mentioning that scrap addition can affect steel quality and product lines.
9. Towards the end of the caption to Figure 3 (lines 254-255), I believe that “upper left corner” and “lower right corner” should be switched.
10. The section beginning at line 264 should be checked for the use of English. I have not seen the term “unmarked carbon tax” before (line 277) and it returns zero results in a web search. It’s not clear what an “uncompromising” islanded energy system could be (line 308).
11. You should be careful to ensure that discussions of the future are put in a way that makes clear that the results are projections, and that these are things that could happen if certain other projections (/ government strategies, etc.) come to pass. For example, line 289 (on Iran’s reliance on gas-powered electricity in 2050) is written in the past tense and there is no reference to the fact that the results are based on projections.
12. The section on ML results (starting on line 332) comes in quite strongly with ML terminology, with little explanation – this could be improved to make the text easier to read for those without experience of ML.
13. Line 412 refers to Australia needing >4 times the grid capacity, but it’s not clear what this figure is for – is it for the 50 % technology diffusion in 2050 scenario?
14. Line 438: should be (iii), not (ii).
15. Line 469 contains “(ref)”, which I assume should be replaced or removed.
16. On line 724 it is stated that the ML algorithm learns from 675 data points: 44 regions, 5 RE data years, 3 installation years. But $44 * 5 * 3 = 660$. Were there actually 45 regions? In any case, this could be clearer.
17. Line 726 refers to Section 4.2, however the sections are not numbered. I suggest ensuring that any references to sections are clear.

Response to Reviewer Comments

We are very grateful for the constructive and detailed comments by all the reviewers. These comments have been carefully considered and addressed; our point-to-point response is presented below.

Notes of other changes: (i) the abstract shortened to meet the 150 words limit; (ii) sub-section titles shortened to meet the length limit; (iii) subsection titles removed from the Discussion section as per instruction; (iv) the subsection "Production system feasibility at-scale" relocated from Discussion to (the end of) Results, as it actually contains new results (i.e. Table 1) and the relocation also makes the Discussion section easier to read after removing the subsection titles; and (v) the Code availability section revised to include online links to codes and their documentation.

Reviewer	Comment #	Comment	Subject	Response
#2	1	An overarching comment is that the authors should add some caveats concerning uncertainties in the assumptions and discuss the results against those uncertainties. I think the results are pretty robust but things could shift around a little with different assumptions and input data.	Assumptions	The uncertainties and caveats that inevitably come with modelling have been further stressed throughout the manuscript, especially in regard to the assumptions made around future energy systems, electricity prices and globally traded commodity prices. Specific reviewer comments that fall under this heading have been individually responded to.
#2	2	Another overarching comment is that it is not clear what the machine learning approach actually does or contributes (it is described in language that most readers cannot follow). I would also recommend that the authors go through the text checking that all information given is relevant and important as to me there seems to be unnecessary detail and information.	ML	ML is discussed in two main sections of the report - Results (alongside Figure 5) and Methods - which have both been edited to improve readability and conciseness. The text has been restructured to ensure more effective communication of the importance of using ML in our study.
#2	3	Line 11 and 34: You may wish to consult the latest IPCC report for a number on share of emissions	Lit Review	We have done the check but this data does not seem to be available in the latest IPCC Report (https://report.ipcc.ch/ar6/wg2/IPCC_AR6_WGII_FullReport.pdf). Therefore, we did not make a change here, noting that other recently published papers have similarly referenced the IEA Report (for example, https://www.nature.com/articles/s41558-022-01383-9).
#2	4	Line 19: The assumptions/discussions around islanded versus grid connected are problematic, see below.	Assumptions : electricity	This issue has been addressed in comments #11, #19, #26 and #32.
#2	5	Line 31: present electricity demand also as TWh	Data	Amended to include TWh: "...and 5 EJ (1400 TWh) of electricity..."
#2	6	Line 39: Avoid using the term "hard to abate" since it is misleading. Is steel harder than anything else? It seems pretty easy through the HDRI-route	Language/ty po	The authors agree with this sentiment, and hence have replaced "hard-to-abate" with "emission-intensive".

#2	7	Line 41-42: Many would disagree that materials efficiency measures have already been adopted. I think steel is heavily overused and would look for references by Julian Allwood and Jonathan Cullen on this. Among options I think many steelmakers are also considering smelt reduction with BOF to handle lower grade ores. Better check this option.	Lit Review	The issue in the original phrase was the grouping of material and energy efficiency. Now, material efficiency has now been dealt separately to energy efficiency, and relevant references (including the IPCC Report) added to support arguments. Please refer to the Introduction (end of first and start of second paragraph). A reference to smelting reduction has been added to the technology landscape within the introduction. SR-BOF is similar to the BF-BOF in that they rely on fossil fuels and require successful CCUS installations for decarbonisation. SR-BOF, however, only accounts for 0.4% of current production whereas BF-BOF accounts for 72% (https://www.irena.org/-/media/Files/IRENA/Agency/Events/2022/Mar/IID22_Canada_Day-2_S4-and-closing.pdf?la=en&hash=BAA30CE010EA89510A062997BB540C8338470D08).
#2	8	Line 69: Since beneficiation is so important please consider adding some explanation/detail on this process.	Ore beneficiation	Additional detail on beneficiation (which originally was in the Method section) has been added to the Introduction: "Beneficiation, which describes the increases ore Fe-content through physical and/or chemical separation processes to remove impurities (commonly silicon, aluminium, phosphorus, and sulphur) from, and thereby increase the Fe-content of, iron ore, is increasingly relied on by the iron ore industry to accommodate lower grades of mined ore."
#2	9	Line 75 onwards: You should take more care in finding and referencing earlier work on this topic. See for example Gielen et al https://doi.org/10.1111/jiec.12997 and Trollip et al https://doi.org/10.1080/14693062.2021.2024123 . A proper search may reveal more.	Lit Review	The two suggested sources, in addition to the Start with Steel Report from the Grattan Institute, have been added into the Introduction. Refer to sentence starting with: "Industrial relocation as an enabler for industrial decarbonisation..."
#2	10	Line 104: What does it add do different levels of scrap? Is this important to keep?	Assumptions : scrap	The different levels of scrap charging to EAF (0%, 25%, 50%) have been included as an auxiliary analysis of LCOS which provides insights into the effect of scrap on future green H2-based steel markets, and hence important to keep. As scrap level increases, the SF load and EAF energy consumption both decrease, hence improving the economics of green H2-based steelmaking. In the Results, the paragraph starting with "EAF scrap charging can drive cost benefits, so long as contaminants are controlled to not affect steel quality and product lines," explains the effects in more detail.

#2	11	Line 116-117: Even grid connected mills can have 100 % renewable electricity (even if the total grid has emissions) and the price can be set through power purchase agreements rather than follow the tariff or the spot price if there's a power exchange. The distinction made between islanded and grid connected to me is very misleading. This is more about the details and peculiarities around electricity market regulation and design in different countries. You could keep it roughly as it is but make very clear to the reader that you make a very "artificial" assumption around grid connected electricity.	Assumptions : electricity	If by "grid", one refers to the national grid, the carbon intensity of power supply is inevitably affected by the power mix of the grid. However, if one talks about a local grid run by an RE provider which can be accessed by a steel maker via say a PPA, then this case would be similar to the islanded option, at least physically (as opposed to business-wise). The assumed nature of the islanded option is to represent a power supply scheme which is guaranteed to be 100% RE but need to deal with storage requirements completely on its own. On the other hand, the "grid" option serves to represent a scheme where continuous power supply is obtained by accessing the national power grid and hence subject to the existing power mix. The purpose of making this distinction is to understand the cost and carbon implications. We recognise the limitations, which are detailed in the updated Discussion. For each country, we now also offer a calculated comparable grid power cost that yields the same LCOS as the islanded option, which we hope will provide something more objective to help assess the future direction of travel required for the grid electricity cost (in the final paragraph of the 3rd section in Results): "If a renewables-dominate grid could be secured to power H2-DRI-EAF steel production, an average global electricity price of \$80, \$70 and \$60/MWh (with the lowest being \$62, \$54, and \$46 \$/MWh) would be required in 2030, 2040 and 2050, respectively, to equalise the LCOS (without carbon taxes) across both islanded and grid systems." Besides, the introduction of grid-powered production (in the 2nd last paragraph of the introduction text) has been updated to stress the assumptions made: "For comparison, we also assessed H2-DRI-EAF plants powered by grid electricity, which offers stable energy supply. We assumed that the grid power's carbon footprint was dependent on the forecast power mix and that electricity was charged according to current industrial tariffs, for the sake of assessing conditions for competitive steelmaking and providing recommendations for electricity market reform."
----	----	---	----------------------------------	---

#2	12	Figure 2b: If I am doing the math right this figure implies LCOH at 2-3 USD/kg in 2050. To me this is much too high when many in the business have 1.5 USD/kg as a target. LCOE at 50 USD in 2050 for Sweden is very high. New wind in northern Sweden today is 25-30 USD/MWh.	Data	To clarify the data in Figure 2b, the phrase has been added (right below Fig 2): "By 2050, the LCOS range dropped to \$535-\$831/t, alongside a LCOH2 from \$1.63-2.80/t and LCOE from \$16-50/MWh." LCOE at \$50/MWh in Sweden in 2050 is the output from the optimised RE plant comprised of 24% solar and 76% wind capacity using the global constant wind turbine unit cost of \$835/kW and solar panel unit cost of \$327/kW, and RenewablesNinja solar and wind potential profiles at the iron ore mine. The \$25-30/MWh figure given for wind power in Northern Sweden today would most likely reflect power generation in ideal locations at ideal times (therefore higher capacity factors), perhaps with more competitive wind turbine procurement costs given historical government subsidy schemes. Also note that our LCOE calculated was based on energy consumed by steel making, not by energy generated by the RE facility.
#2	13	Line 172: \$4280 is a typo?	Language/typo	\$4280 was a typo, now corrected to \$428
#2	14	Line 173-175: What is the relevance of general coal prices? The authors should look at the prices of coking coal (important for the economics of BF-BOF)	Data	Clarification of type of coal added (metallurgical coal), and revision of text to exclude reference to thermal coal.
#2	15	Line 197-199: I question the importance or relevance of including scrap shares in the analysis, and in any case scrap prices may change a lot over time.	Assumptions : scrap	Please refer to comment #10 for why we consider the inclusion of scrap levels as a useful analysis. Yes, scrap steel prices are likely to change over time. Nevertheless, we think that current scrap prices at national-level provide good insights into material value relative to the global market and therefore offer a useful differentiation between countries.
#2	16	Line 210-212: LCOE is a key assumption that is worth discussing more. It is quite likely that LCOE (and LCOH) will much lower than assumed here.	Assumptions : electricity	Please note that the adopted LCOE and LCOH2 were not our assumptions but results of our (LCOS-minimising) modelling. To stress this and to recognise the future potential of lowering these costs, we have added a statement (2nd last paragraph, page 7): "Although the LCOE and LCOH2 were determined by the cost-minimising model, it is likely that they will be cheaper given optimal location of solar and wind plants (i.e., not at the iron ore mine itself); the global weighted average LCOE of new solar PV and onshore wind projects in 2021 were \$48/MWh and \$33/MWh, respectively."
#2	17	Line 219-223: What is the source of labour costs and are the wages representative of skilled steelworkers?	Data: labour	Please refer to Table S18 - steelworker wages are 30% above national wages, with tax added on top. Method has been adjusted for accuracy, and consequently results have shifted slightly with changing shares of labour costs per tonne of steel.
#2	18	Line 256-263: I think these results should be presented as indicative and	Assumptions : electricity	The text following Figure 3 has been adjusted to emphasise uncertainties and caveats.

		with caveats. For example, blue hydrogen may be an option for Russia.		
#2	19	Line 264-265: I think this whole section is misleading and it is mixing cost (in islanded systems) with price in grid systems. Even with grid connection you can have 100 % wind and PV through PPAs. This needs to be rewritten or at least highlight that the distinction made is quite artificial.	Assumptions : electricity	Please refer to comment #11.
#2	20	Line 332-333: This section can do a better job of explaining what the ML does.	ML	ML is discussed in 2 main sections of the report - Results (alongside Figure 5) and Methods - which have both been edited to improve readability and conciseness, and to communicate more effectively the importance of using ML.
#2	21	Line 393: Do you consider that water is or can be recycled after the DR-step?	Assumptions : water	Yes, the assumption is that 12 L/kg H2 is required for electrolysis (above stoichiometric minimum of 9 L/kg H2) however 9 L/kg H2 is returned from iron ore reduction. Therefore, net water requirement of 3 L/kg H2.
#2	22	Line 398-405: Why is size of ore reserve a hard constraint when ore is easily traded? What will be the LCOE of nuclear? (BTW be careful with language and distinguish between steel export and HBI export)	Assumptions : trade	This study focusses on full H2-DRI-EAF steel production at iron ore mine (therefore no ore or HBI trading was considered), and is a baseline analysis for future supply chain modelling.
#2	23	Line 410: "expansive" should be "extensive"?	Language/typo	"expansive" now replaced by "extensive"
#2	24	Line 423-424: What does it mean that investment has been "poured"?	Language/typo	Hyperbole was used to describe the large quantity of green steel investment in Sweden. For clarity, "poured" has now been replaced by "directed".
#2	25	Line 442: Ore can also be traded/exported?	Assumptions : trade	Yes, but it's outside the scope of this baseline study (see comment 22)
#2	26	Line 492-493: Here you make an important observation that basically undermines much of your discussion/results on grid connected production.	Assumptions : electricity	We make the comment here regarding grid-powered systems that "steelmaking facilities may be able to take advantage of dynamic pricing, where production is aligned with demand troughs..." This observation has been included to contextualise the assumptions on electricity tariffs. Treatment of islanded/grid energy system comparisons has now been refined in the Introduction and Results sections (see comment 11), to avoid undermining of results.
#2	27	Line 507: of annual global what?	Language/typo	Adjusted to "annual global iron ore output"
#2	28	Line 530: There's a paper by Vogl on this: https://doi.org/10.1016/j.joule.2021.09.007	Lit Review	Comment removed regarding future work. Vogl's work is now referenced in the Introduction.

#2	29	Line 549-560: How much electricity storage at what cost is used in islanded systems? I imagine an islanded system with lots of H2 storage to operate 24/7 and some electricity storage to run pumps, motors, etc. but what roughly is the share of each?	Energy storage	Figure 2 has now been updated so that costs for 'energy storage' (CGH2, batteries, FCs) are separated from costs for 'production plant' (2b compressor, DR furnace, EAF, casters); previously they were all grouped under 'production facilities'. Method has also been updated so that equations deal with production and storage-related infrastructure separately. In addition, details on the distribution of energy storage costs have now been included in the text following the figure: "Across all cases most storage costs were allocated to CGH2 (mean 91%) with some electricity storage in batteries to manage RE variability; on average, 50% of produced H2 was stored temporarily as CGH2."
#2	30	581-588: I wonder if this level of detail is needed and relevant for the aims of the paper.	Language/typo	We agree that for researchers working on steelmaking, these details are not needed. However, considering that this is the Method part and for the benefit of those who are not familiar with the process, we have opted to keep the description of the steelmaking process.
#2	31	Line 592: word missing?	Language/typo	"not" added
#2	32	Line 685: As noted already current industrial electricity tariffs are not very relevant. Also, it is very risky to mix different sources as in S13.	Assumptions : electricity	Please see comment 11 - Industrial electricity tariffs have been nominally used to assess the required movement of grid energy mixes and prices to support renewables-based production systems. We agree that the previous use of multiple sources was risky. For improvement, industrial electricity tariff data has been updated so that only two sources are used (primarily the IEA dataset); although we would like to use a single source, data gaps (i.e. missing data for certain countries) forced us to seek two datasets. Please refer to Table S13 for revised values.
#2	33	Line 715: This section is rather incomprehensible to me. Would it be possible to explain the ML in more layman's terms?	ML	ML is discussed in 2 main sections of the report - Results (alongside Figure 5) and Methods - which have both been edited to improve readability and conciseness, and to communicate more effectively the importance of using ML.
#2	34	I have not scrutinized the supplementary material but note that some tables are missing sources/references.	Data	All data tables within Supplementary Methods have been reviewed to ensure accurate referencing.
#3	35	Line 67 – “Distinct from” instead of “distinct to”?	Language/typo	"distinct to" now replaced by "distinct from"
#3	36	Line 172 Typo on BFBOF steel costs, \$4280-\$547/t	Language/typo	\$4280 was a typo, now corrected to \$428
#3	37	Line 592 “...NOT yet been extracted”..	Language/typo	"not" added
#3	38	Line 606 Explain “dmt”.	Language/typo	dmt refers to dry metric tonne, which has now been defined in the first instance of use

#3	39	Line 706 “THAT removes ...”, or “while removing carbon from direct reduction of iron”	Language/typo	"that" added
#3	40	Line 35 – Bataille et al 2021 had three demand forecasts: low (1.9 Gt/yr in 2050, evolving to 200 kg/cap/yr in 2080; 2.2 Gt/Yr 250”; and 2.5 Gt/yr 300”). You are likely referring to the higher value.	Data	Data has been updated to cite the 'medium' forecast: "...projection to increase to 2.19 billion tonnes by 2050 given global demand evolving to 250 kg/capita (Bataille et al., 2021)"
#3	41	Line 40 – You may wish to distinguish iron ore reduction from steel making here, but perhaps you do this later.	Language/typo	Distinction between reduction and steelmaking has been added to the end of this sentence: "...during which emission-intensive carbon-based iron ore reduction occurs (producing metallic iron to feed into the steelmaking furnace)."
#3	42	Line 70 – It may or may not be useful for your paper that Bataille et al 2021 geographically structured its scenarios based on <= 100, 200 and 300 km to known feasible CCS reservoirs, and allocation of HDRI stock to a minimum of 3.5 watts per m ² per day solar PV using global databases and the locations of existing facilities or a 1-0 national switch denoting access to inexpensive hydropower.	Lit Review	Thanks for sharing these. We feel that these details are not directly relevant to the modelling carried out in this work so no change was introduced.
#3	43	Line 69-71, Line 159-164 (Kazakhstan) I will be looking to see how beneficiation/ore upgrading is treated	Ore beneficiation	Details on ore beneficiation cost calculations are included under the section titled 'Location selection and geospatial characterisation' in Methods, and limitations detailed in the Discussion.
#3	44	Lines 220-223 I find it hard to believe wages in Canada (\$43/hr) are more than Sweden (\$35/hr). You may wish to check this, and how you accounted for relative income tax costs.	Data: labour	Please refer to Table S18 - steelworker wages are 30% above national wages, with income tax added on top. Method has been adjusted for accuracy, and consequently results have shifted slightly with changing shares of labour costs per tonne of steel.
#3	45	Line 358-375 (Figure B), also Line 473-477. Given the progressive degradation of the Pilbara ores, you should say something about what the huge dark green dot in Western Australia, seemingly the size of the rest put together. Is it beneficiated magnetite, goethite, a mix ?	Ore beneficiation	The phrase has been added in the ML Results section: "A striking opportunity exists in Western Australia, a region offering a stable investment environment, so long as beneficiation can manage the progressive degradation of Pilbara ores." The Discussion further details the ore beneficiation challenge in the face of decreasing mined ore quality, and limitations in the present ore cost modelling (see comment #50).
#3	46	Line 434 What is meant by water “abstractions”? Total use? Net loss after recycling? Subtractions?	Assumptions : water	Table 1 has now been updated with “net water demand” as opposed to “water abstractions”. Previously the data referred to gross water demand (i.e. not taking into account recycled water in DR step), but now the recycled water loops are included.
#3	47	Line 434 Gross iron in ore reserves (over a certain concentration?) may be a useful value	Data	Gross iron content remaining in reserves (Mt) has been added to Table 1.

#3	48	In line 434 I would think carefully about what to add and what to subtract.	Data	We are not exactly sure what this comment refers to, but believe it is in reference to the use of (-) in the scrap surplus column of Table 1. If the scrap demand is greater than scrap availability, we indicated such with a negative value (implying a negative surplus, that is, a deficit). This is now noted in the caption of the table.
#3	49	Line 451-464 - Correct me if I am wrong, but all your analysis focuses on trade route where the full H-DRI-EAF process is done in one place, not where sponge or HBI is shipped to EAFs in the target market e.g., (Bataille et al., 2021; Trollip et al., 2022). There are tremendous number of different final steel types and chemical mixes, whereas reduced iron is moved and used in only a few ways, arguably making shipping of reduced iron for final processing in a BOF or EAF easier for final industrial structure to absorb.	Assumptions : trade	We certainly agree with this comment that transnational supply chain analysis is required to fully value green H2-based steel production, and we are exploring this in a subsequent study. The current study, however, focusses on full H2-DRI-EAF steel production at iron ore mine, which has not considered sponge or HBI trading.
#3	50	Line 473 – This is where you will receive your most reader push back, as there is no discussion of differential beneficiation, which differs tremendously by ore type. Arguably, this is the weakest part of the paper, but you do seem to at least acknowledge this from line 500 onward.	Ore beneficiation	Further clarification of limitations in ore modelling has been included in the Discussion: "...characterisation of ore deposits according to type composition (hematite, magnetite, goethite, limonite, etc.) was scarce, and consequently the assumption made that all ore reserves were hematite (the most common iron ore type). Given better data, we would have performed differential beneficiation and reduction analysis to determine more accurate production costs."
#3	51	Line 487-493. Industrial facilities DO NOT want to align production to minute by minute, hourly RE availability – that is simply not bankable/investable. However, they can size energy resources and storage to their 24/7 needs, and build that into the business plan (Trollip et al., 2022). For example, Hybrit is building a cavern to store several days of necessary hydrogen, and is planning to arbitrage North Sea wind excesses/low price periods.	Assumptions : electricity	Our optimisation model does exactly what is suggested to be practical here, i.e. the employment of energy storage to mediate RE supply and steel making energy demand; the assumed production-side flexibility was limited to allowing the electrolyser and EAF loads to vary while keeping SF's load constant.
#3	52	Lines 494-499 I encourage you to investigate this further in future papers.		Yes, the intention is to look at other flexible energy-production system optimisations in future papers.
#3	53	Line 525 – Do you have a reference for the different melting points of reduced iron by carbon content?	Data	Given reference should sit at the end of this sentence, now it reads: "...most notably the maintenance of metallisation degree in the shaft furnace in light of the resistance of diffusion of H2, and the different melting characteristics of carbon-lean H2-DRI in the EAF (Kim & Sohn, 2022)."
#3	54	Figure 6 Have you included a valuation of the separated oxygen?	Assumptions	The sale of oxygen is not considered given the variance in global oxygen markets and limited

				contribution of oxygen credits to overall steel cost (e.g. oxygen revenues of 14.9 EUR/tLS were calculated by Vogl et al. (2018)).
#3	55	Lines 635-637. The assumption of maximum 50% scrap may hold globally or nationally, but not on a facility basis and is not justified. It should have been set based on regional availability – scrap availability is heavily dependent on level and stage of development, and the possibility of import.	Assumptions : scrap	It is understood that ideally the scrap availability should ideally be set based on regional scrap flows, including import/exports, however this was outside of the study's scope. This comment has been added to the Discussion: "Ideally, scrap charge to the EAF should be set based on regional scrap flows, including import and export."
#3	56	Line 664 You use an 8% discount rate in the CRF here, and 7% previously (Line 628)? Is this a typo?	Data	7% discount rate was a typo, now corrected to 8%.
#3	57	Line 691-694 it might be useful to include the IEA 2021 carbon prices here for reference.	Data	Added: "...equal to \$250, \$200 and \$55/t CO2 in 2050, respectively (refer to Table S16)."
#4	58	A linear relationship is used to calculate DR pellet cost calculation based on iron content. To what extent is this a good fit to observed cost data?	Data	We would like to point out that the observed cost data did not offer any clear relationships, which seems to be understandable since beneficiation costs are highly dependent on specific mineral types. In the absence of a justifiable more sophisticated model, we have adopted the simplified linear relationship, which is illustrative of the added cost burden, but not conclusive.
#4	59	In Fig. 2b, the cost uncertainty for Kazakhstan is considerably greater than the other countries, and should possibly be addressed further in the text.	Data	Kazakhstan's cost uncertainty surrounding ore beneficiation is addressed in the text following Figure 2b: "In Kazakhstan, where..."
#4	60	Generally the work assumes that green steel will be produced using wind and solar power, and little mention is given to hydropower and nuclear power. It may be worthwhile addressing how the results might change if these other sources are considered. This could be particularly interesting for Sweden, which appears to have the highest LCOE of the 17 countries shown in Fig. 2b but which currently meets the vast majority of its electricity demand using hydro and nuclear.	Assumptions : electricity	Multiple references have now been made to hydropower: Bottom of page 9: "In contrast, grid-based systems in Canada and Sweden (closely followed by Brazil) may emit just 0.2 t CO2/t steel in 2030 (see Figure S3b) due to their grid portfolios with substantial hydropower and nuclear shares." Last paragraph of page 10: "Whilst we investigated solar and wind resources, other stable and clean electricity sources (i.e. hydropower) are ideally placed for electricity-intensive steel production."
#4	61	It may be worth briefly mentioning energy storage costs in the results following Fig. 2, as they are currently not shown in the cost breakdown of Fig. 2b or mentioned in the text following the figure.	Energy storage	Figure 2 has now been updated so that costs for 'energy storage' (CGH2, batteries, FCs) are separated from costs for 'production plant' (2b compressor, DR furnace, EAF, casters); previously they were all grouped under 'production facilities'. Details on the distribution of energy storage costs have now been included in the text following the figure: "Across all cases most storage costs were allocated to CGH2 (mean 91%) with some electricity storage in batteries to manage RE variability; on average, 50% of produced H2 was stored temporarily as CGH2."

#4	62	On line 138, I would suggest that “levelised costs of green steel (LCOS)” might make more sense than “green levelised costs of steel (LCOS)”.	Language/ty po	changed to "levelised costs of green steel (LCOS)"
#4	63	On line 146, the costs from the IRENA report should be per kW, not per MW.	Language/ty po	Adjusted to "/kW"
#4	64	I believe there is a rogue digit in one of the costs on line 172 (currently says \$4280-\$547).	Language/ty po	\$4280 was a typo, now corrected to \$428
#4	65	In or around the paragraph starting on line 191, it may be worth mentioning that scrap addition can affect steel quality and product lines.	Language/ty po	Topic sentence added at the start of this paragraph to reiterate the need to control contaminants in scrap steel inputs: "EAF scrap charging can drive cost benefits, so long as contaminants are controlled to not affect steel quality and product lines. Figure 3a shows that scrap addition generally reduces the LCOS, however benefits lessen over time with cheaper renewable energy..."
#4	66	Towards the end of the caption to Figure 3 (lines 254-255), I believe that “upper left corner” and “lower right corner” should be switched.	Language/ty po	“upper left corner” and “lower right corner” have now been switched
#4	67	The section beginning at line 264 should be checked for the use of English. I have not seen the term “unmarked carbon tax” before (line 277) and it returns zero results in a web search. It’s not clear what an “uncompromising” islanded energy system could be (line 308).	Language/ty po	"unmarked" replaced by "marginal" "uncompromising" replaced by "independent"
#4	68	You should be careful to ensure that discussions of the future are put in a way that makes clear that the results are projections, and that these are things that could happen if certain other projections (/ government strategies, etc.) come to pass. For example, line 289 (on Iran’s reliance on gas-powered electricity in 2050) is written in the past tense and there is no reference to the fact that the results are based on projections.	Language/ty po	Language adjusted to emphasise uncertainties throughout the discussion, for example, "...Iran’s projected reliance on gas-powered electricity meant that even by 2050, forecasted grid emission-intensity is greater than the BF-BOF route"
#4	69	The section on ML results (starting on line 332) comes in quite strongly with ML terminology, with little explanation – this could be improved to make the text easier to read for those without experience of ML.	ML	ML is discussed in 2 main sections of the report - Results (alongside Figure 5) and Methods - which have both been edited to improve readability and conciseness, and to communicate more effectively the importance of using ML.
#4	70	Line 412 refers to Australia needing >4 times the grid capacity, but it’s not clear what this figure is for – is it for the 50 % technology diffusion in 2050 scenario?	Data	Clarification added (1st paragraph, page 14): "...given that over 4 times the national energy grid capacity would be required in 2050 (assuming 60% technology diffusion)..."
#4	71	Line 438: should be (iii), not (ii).	Language/ty po	corrected to "(iii)"

#4	72	Line 469 contains "(ref)", which I assume should be replaced or removed.	Language/typo	"(ref)" removed
#4	73	On line 724 it is stated that the ML algorithm learns from 675 data points: 44 regions, 5 RE data years, 3 installation years. But $44 * 5 * 3 = 660$. Were there actually 45 regions? In any case, this could be clearer.	ML	This has now been clarified in the Methods section: "The ML algorithm learned from 675 data points: 45 regions modelled over 5 renewable energy input data years and 3 installation years. Note that New Zealand was the 45th region added to the 44 previously optimised regions to ensure the largest range of latitudes were covered in the input dataset."
#4	74	Line 726 refers to Section 4.2, however the sections are not numbered. I suggest ensuring that any references to sections are clear.	Language/typo	Apology - this was an old reference from a previous version, now corrected to "Equation 1".

REVIEWERS' COMMENTS

Reviewer #3 (Remarks to the Author):

Summary

- The authors did an exemplary job answering the reviewers. I will leave it to the Nature editors to assess the paper but if the following points are clarified this is ready to go.

Substantive

- In Figure 3, is the “increased competitiveness” in the bottom right supposed to be decreased competitiveness?? Given #1 is cheapest? I struggled with #1 being cheapest on both the HDRIEAF and BFBOF dimensions, and I imagine others will too ... What are the # values on the axes?? The US, Canada and Sweden are most costly per tonne, so aren't they less competitive??? Ok, now I think I understand, a country is in the bottom right if the transition from BFBOF to HDRIEAF helps them the most relatively. I would maybe consider some alternate labelling. Maybe remove the "decreased" and "increased competitiveness", and just say there further right and down, the more a country benefits by the BF-BOF to HDRIEAF transition.
- Are the values for the axes for Figure 5a correct, at \$50-\$300/USD tonne steel? Isn't this for the renewables costs? I suspect “USD/tonne steel” is correct for panel (b), but not (a). Maybe "energy costs USD per tonne steel"?

Reviewer #4 (Remarks to the Author):

I am happy that all of my comments have been addressed, though I think that the text around Fig. 2b could still be improved. It looks like Kazakhstan is now Canada (hopefully this is correct!) and the error bar for Canada is very large but no explanation is given for this.

Response to Reviewer Comments

We are very grateful for the further checking by all the reviewers. Their comments have been carefully considered and addressed; our point-to-point response is presented below, and the changes to the manuscript (all in figure or table captions, including both responses to comments and our own minor edits) are highlighted.

REVIEWERS' COMMENTS

Reviewer #3 (Remarks to the Author):

Summary

- The authors did an exemplary job answering the reviewers. I will leave it to the Nature editors to assess the paper but if the following points are clarified this is ready to go.

Response: We are very grateful for your checking of our revised manuscript.

Substantive

- In Figure 3, is the “increased competitiveness” in the bottom right supposed to be decreased competitiveness?? Given #1 is cheapest? I struggled with #1 being cheapest on both the HDRIEAF and BFBOF dimensions, and I imagine others will too ... What are the # values on the axes?? The US, Canada and Sweden are most costly per tonne, so aren't they less competitive??? Ok, now I think I understand, a country is in the bottom right if the transition from BFBOF to HDRIEAF helps them the most relatively. I would maybe consider some alternate labelling. Maybe remove the "decreased" and "increased competitiveness", and just say there further right and down, the more a country benefits by the BF-BOF to HDRIEAF transition.

Response: Many thanks for the suggestions. Figure 3 has been altered so that "decreased" and "increased competitiveness" have been removed and the figure legend updated to include “The closer the country is placed to the bottom right-hand corner, the more it benefits from the BF-BOF to green H2-DRI-EAF transition.” We believe the revised figure is now much easier to follow.

- Are the values for the axes for Figure 5a correct, at \$50-\$300/USD tonne steel? Isn't this for the renewables costs? I suspect “USD/tonne steel” is correct for panel (b), but not (a). Maybe "energy costs USD per tonne steel"?

Response: Thanks for spotting the error. Figure 5a has been updated so that the axis label is now “energy costs USD per tonne steel”.

Reviewer #4 (Remarks to the Author):

I am happy that all of my comments have been addressed, though I think that the text around Fig. 2b could still be improved. It looks like Kazakhstan is now Canada (hopefully this is correct!) and the error bar for Canada is very large but no explanation is given for this.

Response: Many thanks. We can confirm that the descriptions regarding Kazakhstan and Canada in the figure and the related text are correct. Figure 2b legend has been improved, including an explanation for Canada's large error bar: “*Canada has an especially large error*”

bar due to the variance in solar and wind energy across the large land mass where various iron ore deposits are distributed'.